DOI: 10.1038/s41467-018-05079-7　　**OPEN**

# Red blood cell-hitchhiking boosts delivery of nanocarriers to chosen organs by orders of magnitude

Jacob S. Brenner [1,2], Daniel C. Pan[2], Jacob W. Myerson [2], Oscar A. Marcos-Contreras[2], Carlos H. Villa [2,3], Priyal Patel[1,2], Hugh Hekierski[4], Shampa Chatterjee[5], Jian-Qin Tao[5], Hamideh Parhiz[2], Kartik Bhamidipati[2], Thomas G. Uhler[2], Elizabeth D. Hood[2], Raisa Yu. Kiseleva [2], Vladimir S. Shuvaev[2], Tea Shuvaeva[2], Makan Khoshnejad[2], Ian Johnston[2], Jason V. Gregory[6], Joerg Lahann[6], Tao Wang [7], Edward Cantu[8], William M. Armstead[4], Samir Mitragotri[9] & Vladimir Muzykantov[2]

Drug delivery by nanocarriers (NCs) has long been stymied by dominant liver uptake and limited target organ deposition, even when NCs are targeted using affinity moieties. Here we report a universal solution: red blood cell (RBC)-hitchhiking (RH), in which NCs adsorbed onto the RBCs transfer from RBCs to the first organ downstream of the intravascular injection. RH improves delivery for a wide range of NCs and even viral vectors. For example, RH injected intravenously increases liposome uptake in the first downstream organ, lungs, by ~40-fold compared with free NCs. Intra-carotid artery injection of RH NCs delivers >10% of the injected NC dose to the brain, ~10× higher than that achieved with affinity moieties. Further, RH works in mice, pigs, and ex vivo human lungs without causing RBC or end-organ toxicities. Thus, RH is a clinically translatable platform technology poised to augment drug delivery in acute lung disease, stroke, and several other diseases.

[1] Pulmonary, Allergy, & Critical Care Division, Department of Medicine, University of Pennsylvania, Philadelphia, PA 19104, USA. [2] Department of Systems Pharmacology and Translational Therapeutics and Center for Translational Targeted Therapeutics and Nanomedicine, Perelman School of Medicine, University of Pennsylvania, Philadelphia, PA 19104, USA. [3] Division of Transfusion Medicine and Therapeutic Pathology, Department of Pathology, University of Pennsylvania, Philadelphia, PA 19104, USA. [4] Department of Anesthesiology & Critical Care, University of Pennsylvania, Philadelphia, PA 19104, USA. [5] Institute for Environmental Medicine, University of Pennsylvania, Philadelphia, PA 19104, USA. [6] Department of Chemical Engineering and Biointerfaces Institute, University of Michigan, Ann Arbor, MI 48109, USA. [7] Penn Cardiovascular Institute, University of Pennsylvania, Philadelphia, PA 19104, USA. [8] Division of Cardiovascular Surgery, Department of Surgery, University of Pennsylvania, Philadelphia, PA 19104, USA. [9] School of Engineering & Applied Sciences, Harvard University, Wyss Institute, Cambridge, MA 02138, USA. These authors contributed equally: Jacob S. Brenner, Daniel C. Pan, Jacob W. Myerson, Oscar A. Marcos-Contreras. Correspondence and requests for materials should be addressed to J.S.B. (email: brennerj@uphs.upenn.edu) or to V.M. (email: muzykant@mail.med.upenn.edu)

For decades, the field of nanomedicine has held the promise of delivering drugs to specific target organs while avoiding off-target side effects. Yet the theoretical concept of drug targeting, first proposed by Paul Ehrlich more than a century ago (his "magic bullet" theory), has run into numerous practical challenges[1]. Rapidly after injection, nanocarriers (NCs) inherently accumulate to very high levels in the liver and spleen, part of clearance by the reticulo-endothelial system (RES)[2]. Due to this and other unfavorable factors, NCs are not able to accumulate in many organs of therapeutic interest, most notably the heart and brain, organs whose pathologies cause the majority of the world's deaths. For example, current nanomedicine formulations achieve at best 1% of the injected dose (%ID) accumulating in the brain[3]. Affinity moieties such as antibodies have been proposed to increase target accumulation, but they have their own challenges, including unintended effects on host defense (caused for example by IgG multiplexed on NCs), the common necessity of switching affinity moieties between each species, and eventually humanization of affinity ligands. Overcoming these challenges with a technically straightforward and translatable approach for organ targeting of NCs remains the raison d'etre of nanomedicine research.

To achieve this goal, we and others have explored the idea of combining nanomedicine with cell therapy, in particular, red blood cell (RBC) delivery. RBCs are a particularly attractive vascular carrier[4], due to their biocompatibility and clinical safety of transfusion and prior clinical successes using them as drug carriers[5,6]. The engineering of NC–RBC complexes was inspired by the finding that antibodies or bacteria that naturally or artificially adhere to circulating blood cells such as RBCs are removed by specific organs, though usually the liver[7,8]. This provided a biological basis for attempts to deliver NCs to other target organs by shuttling the NCs on the surface of the super-carrier cells, including leukocytes and RBCs[9,10].

Such reasoning led us to previously develop a prototype version of RBC-hitchhiking (RH); nanoparticles (NPs) were mixed with RBCs ex vivo, causing the NPs to adsorb onto the RBCs[11]. The RBC-bound NPs injected intravenously (IV) were serendipitously found to accumulate in the lungs, although to a very modest extent. The hypothesized mechanism was that when RBCs squeeze through the lung capillaries, they transfer the NCs to the pulmonary capillary endothelial cells. This prototype RH suggested an interesting mechanism of transfer, but had key liabilities: the NPs had no drug carrying capacity and caused dose-limiting toxicities to RBCs[12]; and the prototype RH achieved lung delivery of only 3% of the initial dose (ID) of NPs[11], quite modest compared to the 30% ID achieved with lung-targeting affinity moieties identified decades ago, such as anti-PECAM antibodies[13,14]. However, the unique mechanism of delivery suggested a novel path forward for nanomedicine's organ-targeting challenges.

Here we have dramatically improved, generalized, and extended the prototype concept of RH, producing an effective and broadly applicable platform technology. Our goals were to move RH toward being a clinically useful platform technology by finding the optimal NC properties for RH, exploring the mechanism of the RH, determining if RH works in large animals and humans, and evaluating the safety of optimized RH. Perhaps more importantly, we sought to extend RH to not just target the lung, but any organ. In particular, we hypothesized that while IV-injected RH-NCs would accumulate in the lungs (the first capillary bed downstream of an IV injection), intra-arterially (IA) injected RH-NCs would accumulate in the immediately downstream organ. Such IA delivery of RH-NCs could treat two of the leading causes of cardiovascular death: heart attack and stroke. For severe heart attacks and strokes, it is standard of care to insert a catheter into the affected artery to remove flow obstructions, which helps, but still has poor patient outcomes. Therefore, we sought to determine if RH could be injected IA, as an IA catheter would already be in place during heart attack and stroke treatment.

This IA-based extension of RH, if successful, would demonstrate the synergistic power of combining nanomedicine with different size-scale technologies; here, we employed the synergistic combination of "nano-micro-macro": nano-scale drug carriers + micro-scale RBCs + macro-scale IA catheters. The synergy of nano-micro-macro has recently been productively leveraged for other NC and gene therapy delivery challenges, such as intraperitoneally injected NCs that stick to mesothelial cells to act as a drug depot[15], peri-nerve injections of NCs to block pain[16], intratracheally administered NCs that localize near airway epithelial cells[17,18], NCs injected into the brain parenchyma to land on neurons or tumor cells[19–21], and delivery of gene therapy vehicles to overcome similar barriers[22,23]. Building off these prior approaches, the particular nano-micro-macro synergy of RH, if successful, could provide localized, high-concentration drug delivery to any organ into which an intravascular catheter can be clinically deployed.

While the prior nano-micro-macro combinations focus on chronic illnesses, we investigate here RH as a drug delivery platform for acute critical illness. Such illnesses, including acute respiratory distress syndrome (ARDS), pulmonary embolism (PE), and acute ischemic stroke, have poor outcomes with current therapies. These poor outcomes are in large part due to the off-target side effects of systemically delivered drugs, which are particularly intolerable because of the multiple perturbed organ systems of critically ill patients. Additionally, acute critical illness affords unique delivery opportunities, with intravascular catheters commonly used in ways that are impractical for chronic illnesses. Therefore, here we examine RH's delivery properties in models of each of the above critical illnesses, to start down a development path to delivering high concentrations of drugs to the target organs affected by severe, acute pathologies.

## Results

**Effects of NC properties and serum on the absorption of NCs onto RBCs.** The intended clinical workflow of RH is to first adsorb the NCs onto RBCs ex vivo, followed by intravascular injection, which leads to transfer to the target organ (Fig. 1a). Therefore, we began with the first step in the RH process, finding optimal formulations and conditions for NC adsorption onto RBCs. RBCs were isolated from mice by venous blood draw, washed in buffer to remove the serum, mixed with NCs and then incubated for 5 to 60 min, washed, and then analyzed for NC-to-RBC adsorption. Figure 1b shows an electron micrograph of two types of NPs adsorbed on RBCs: polystyrene nanoparticles (PS-NPs) used in the original prototype RH studies and nanogel NCs. Figure 1c shows the RBC adsorption efficiency of a broad range of NCs, including clinically used NCs such as liposomes and albumin NCs.

Notably, free IgG molecules do not exhibit any appreciable adsorption, while all tested NPs do exhibit adsorption, as does the 25-nm gene therapy vector, adeno-associated virus (AAV). Also notable for RH optimization is that the two most mechanically flexible NCs, nanogels and liposomes, show a marked increase in adsorption efficiency when they are surface-coated with a protein, both random IgG (Fig. 1c, d) or albumin (Fig. 1e), which indicates that the identity of the protein coating the NC surface does not appear to be a major factor in adsorption efficiency.

We next tested how RBC–NC adsorption is affected by serum (Fig. 1f), since this factor is highly relevant in the context of

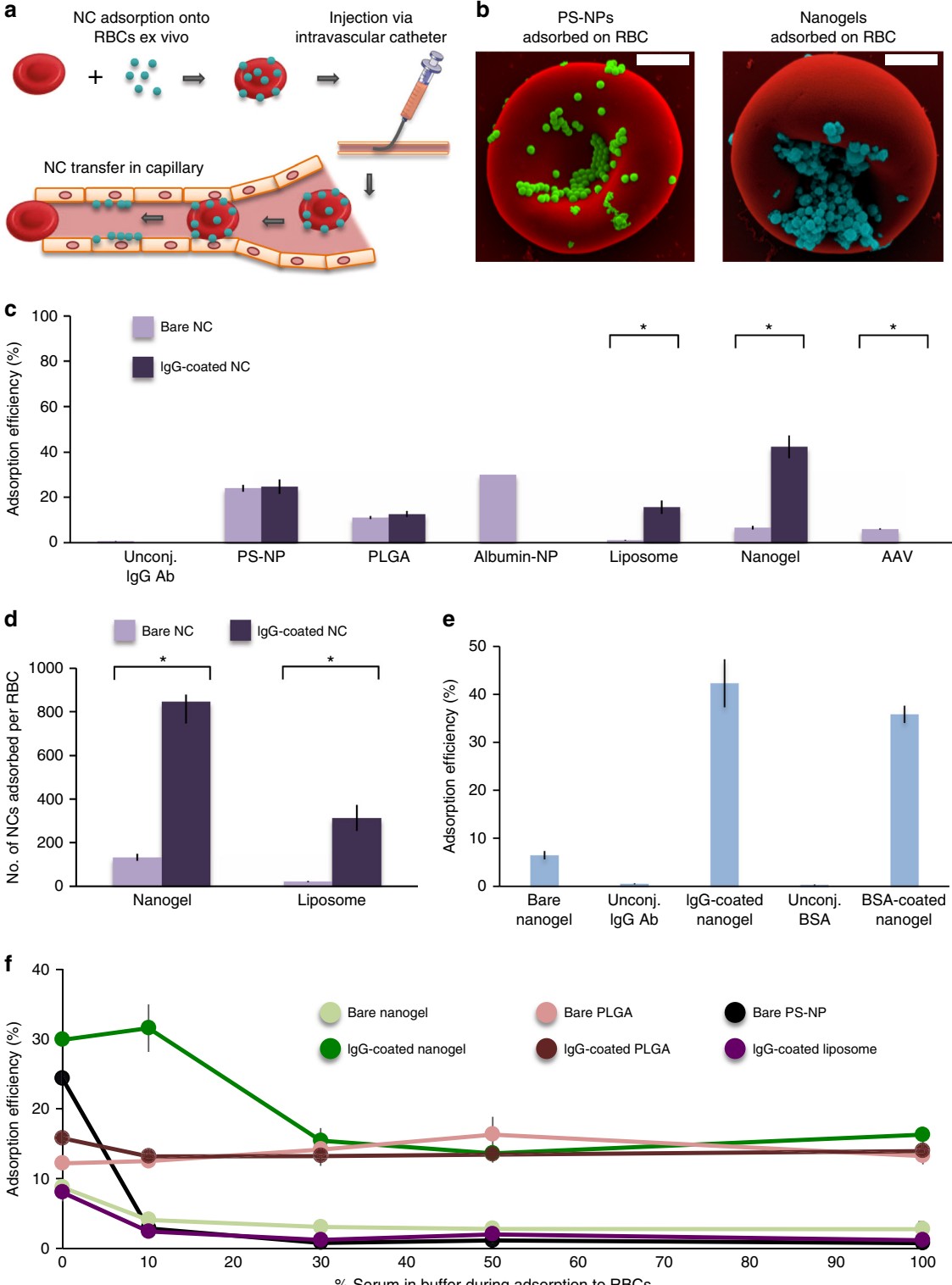

**Fig. 1** Clinically translatable nanocarriers adsorb onto red blood cells. **a** Procedural steps of RBC hitchhiking. NCs are first adsorbed onto the RBCs ex vivo. The RBC–NC complexes are then injected via an intravascular catheter, after which the NCs transfer from the RBCs to the first downstream organ's capillaries. **b** Scanning electron micrographs of PS-NPs and nanogels attached to the surface of murine RBCs. NCs were mixed with RBCs in vitro, leading to adsorption of NCs onto the RBCs. Scale bars = 1 μm. **c** Efficiency of radiolabeled NC adsorption onto RBCs, as defined by the % of total NC added to RBCs that pellet with RBCs. **d** Number of NCs adsorbed per RBC when NCs were mixed with RBCs at a ratio of 2000 NCs per RBC. **e** Adsorption efficiencies onto RBCs of free proteins (IgG and BSA) compared to free nanogels and to nanogels coated with each protein. **f** Adsorption efficiencies of unmodified and IgG-coated NCs with increasing concentrations of serum present in the buffer during adsorption. Each data point represents mean ± s.e.m ($n = 3$). *$P < 0.05$, non-paired, two-tailed $t$-test

loading RBCs for clinical use; packed red blood cell units (PRBCs) used for the vast majority of transfusions typically have ~10% residual serum, while clinically used but less common "washed" PRBCs have <1%. Increasing concentrations of serum unequally affects the adsorption of different NCs onto RBC; serum severely inhibited adsorption of some NCs (e.g., liposomes), but for other NCs, it had only a mild inhibitory effect (IgG-nanogels) or no effect at all (e.g., poly(lactic-co-glycolic acid) NCs [PLGA]).

**NC properties affect RBC transfer to the target organ.** After optimizing NC-to-RBC adsorption, we tested the efficacy of RH

delivery to a target organ in mice to find the optimal NCs for RH. As seen in Fig. 2a, when injected IV, PS-NPs covalently coated with random IgG did not accumulate significantly in the lungs, but RH increased IgG-PS-NP lung uptake almost 20-fold and the lung-to-liver ratio more than 30-fold. Additionally, RH can augment lung uptake of PS-NPs coated with an affinity moiety. We tested PS-NPs coated with anti-PECAM-monoclonal antibodies, as such targeting epitopes have led to higher lung uptake of this type of NP than any other affinity moiety published to date[13,24]. RH increased the lung uptake of anti-PECAM PS-NPs 2.6-fold, with concomitant marked reduction of hepatic uptake,

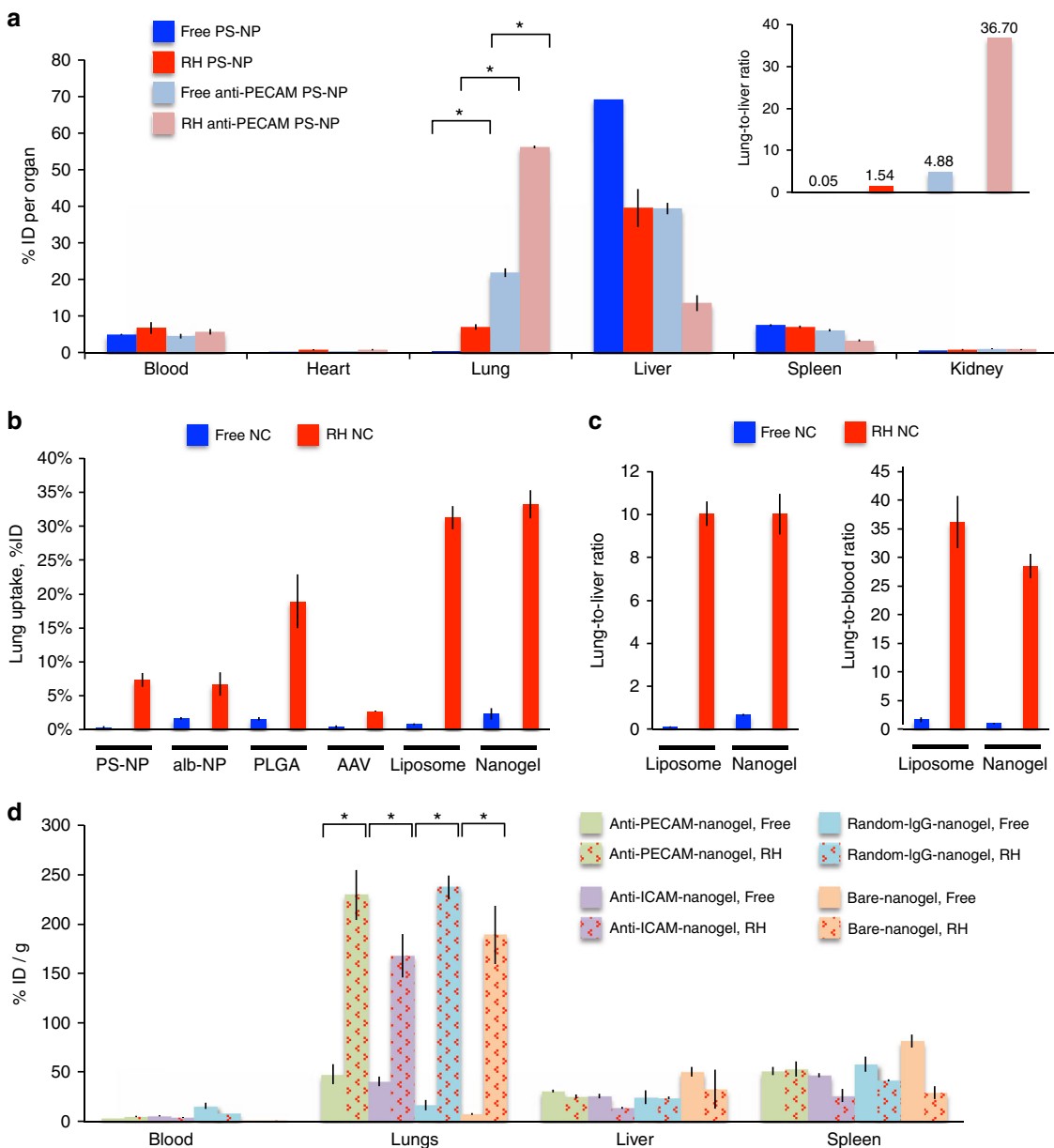

**Fig. 2** IV injection of optimized RH NCs massively augments lung delivery without affinity moieties. **a** Mice were injected IV with polystyrene nanoparticles (PS-NPs) that were covalently coated with either IgG (dark blue) or anti-PECAM antibodies (light blue) and radiolabeled with a trace amount of I-125-IgG. Separate mice were injected with one of those two PS-NPs adsorbed onto RBCs (dark and light red, respectively). Mice were sacrificed 30 min later and I-125 activity in the organs was measured in a gamma counter. Displayed is the percent injected I-125 dose (%ID) for each organ. Inset: lung-to-liver ratios, which are calculated by dividing the lung's %ID per gram of tissue (%ID/g) by the liver's %ID/g. **b** %ID in the lung of different NCs labeled and injected as in **a**. **c** Lung-to-liver (left panel) and lung-to-blood (right panel) ratios of liposomes and nanogels, the two top performing lung-directed RH NCs. **d** Mice were injected with nanogels that were either uncoated (bare) or covalently coated with one of the three different antibodies (random rat IgG, anti-PECAM, or anti-ICAM) (blue bars). Other mice were injected with each of those four antibody–nanogel formulations adsorbed onto RBCs (red bars). Data are plotted as %ID per organ. Each data point represents mean ± s.e.m ($n = 3$). *$P < 0.05$, non-paired, two-tailed $t$-test

resulting in almost an order of magnitude elevation of the lung-to-liver ratio. The combination of affinity moiety plus RH thus increased the lung-to-liver ratio 760-fold over free IgG-PS-NPs (Fig. 2a inset). Thus, RH augments delivery of both untargeted and affinity targeted NCs, and the combined approach has a strong cumulative effect.

We next screened a variety of NPs for their effectiveness in RH. When injected IV, RH augmented lung uptake in the case of every NC tested (Fig. 2b). The fold improvement of lung uptake provided by RH over free NCs ranged from ~4- (albumin-NPs) to ~40-fold (liposomes), with the NCs with the lowest elastic modulus, and liposomes and nanogels having the highest values. Two key variables for determining the effectiveness of a targeting modality are the fold improvements conferred by targeting on the target organ-to-liver and target organ-to-blood ratios. For liposomes, the RH improvements in the lung-to-liver and lung-to-blood ratios were 116× and ~50×, respectively, while RH nanogels achieved values of 15× and 27× (Fig. 2c).

For these high-performing NCs, we next tested how RH interacts with concomitant affinity moieties. Like with PS-NP, RH improves lung targeting of nanogels coated with affinity moieties such as anti-PECAM and anti-ICAM (Fig. 2d). Surprisingly, in RH, nanogels do not require affinity moieties to achieve their maximal binding, lung uptake was equally high whether the nanogels were covered in affinity moieties, random-IgG, or no surface coating at all. Thus, RH nanogels can achieve very high uptake without the use of affinity moieties. Indeed, with no affinity moieties, RH nanogels produce a lung-to-liver ratio 6.4-fold higher than that of the nanogels presenting anti-PECAM antibody, the best known targeting moiety for this size of NPs.

**RH NCs are taken up into cells lining the capillary lumen of target organs**. The hypothesized mechanism underlying RH was originally that NCs on RBCs are transferred to the capillary endothelium, as the RBCs squeeze through the pulmonary capillaries. However, previously, there were no data proving what cell types took up RH NCs, or even that the NCs were taken up by any type of cells at all. Therefore, we IV injected RH fluorescently labeled nanogels and sectioned the lungs for confocal imaging. As seen in Fig. 3a, in naive mice, nanogels do indeed appear to be inside the endothelial cells (VE-cadherin+) and not present in the lumens of large blood vessels, supporting the original hypothesized mechanism.

We next determined if this same RH mechanism is evident in lungs suffering from a pathology that would be a clinically realistic target for RH-based therapies. We focused on the inflammatory disease ARDS, which causes pulmonary capillary endothelial dysfunction, influx of destructive leukocytes, and has been identified as a key target for pulmonary nanomedicine[14]. We used a well-validated mouse model of ARDS, lipopolysaccharide (LPS) instillation into the lungs. As with naive mice, NCs were taken up into the pulmonary capillary endothelial cells. Surprisingly, however, in LPS-challenged animals, the NCs were also taken up by intravascular resident leukocytes (Fig. 3b), localized in the capillary lumen during ARDS and have been implicated in ARDS pathology[25,26]. Thus, RH delivers NCs to multiple different cell types that line the capillary luminal surface, and the cellular specificity of delivery depends on the pathophysiological context.

Having found that intravascular resident leukocytes profoundly take up RH NCs in the LPS-inflamed lungs, we next attempted to directly observe the real time transfer of NCs from RBCs to leukocytes. Intravascular resident leukocytes in ARDS include both pulmonary intravascular macrophages (PIMs) and marginated neutrophils[25–28]. To simulate PIMs, we seeded activated

mouse macrophages onto the luminal surface of microfluidic chambers and exposed them to RH nanogels under flow. In a control setting, activated macrophages did not take up free NCs flowing in the lumen (Fig. 3c). In sharp contrast and good agreement with results obtained in vivo, activated macrophages grabbed NCs flowing at identical concentration in the form of RBC-absorbed NCs, separated the NCs from the carrier RBC, and then released RBCs while retaining the detached NCs (see Supplementary Movies 1 and 2).

**RH does not inflict systemic and local adverse effects**. Having optimized the RH formulation and having determined the basic RH mechanism, we endeavored to validate the safety of RH. We began by checking if the first step of RH, ex vivo adsorption of NCs onto RBCs, causes agglutination of RBCs. RBC agglutination can cause pulmonary emboli, RBC aggregates lodged in the pulmonary arteries. We first assayed RBC aggregation with the clinical protocol of "thin-smear" preparations (Fig. 4a). As described above, PS-NPs, the model NC used in the prototype RH studies[11], causes severe RBC aggregation, on par with that caused by cross-species serum. In sharp contrast, RH-optimized NCs such as nanogels do not cause significant agglutination.

To assess the degree of RBC agglutination, we used the clinical gold standard, the round-bottom-well assay, in which non-aggregated RBCs settle into a tight dot at the bottom of a round-bottom well, while aggregated RBCs, having formed into a film, fail to pack into a tight dot (Fig. 4b). As with thin smears, uncoated PS-NPs cause RBC agglutination at the maximum degree quantifiable by the assay. By contrast, no RBC agglutination is detectable for RBCs adsorbed with any of the PS-NPs covalently coated with protein (as used in Fig. 2), nanogels (± IgG coating), or liposomes. Thus, while some NCs (bare PS-NPs) do cause severe RBC agglutination, most clinically relevant NCs do not cause significant RBC aggregation.

We next tested whether RH causes RBCs to lodge in the target organ. Clogging of the vasculature with RBCs could impede perfusion to the target organ, cause pro-adhesive activation of endothelium and a pro-thrombotic state, and in the case of clogging in the lung, cause increased pulmonary arterial pressures (PAPs) and subsequent right heart failure. To assay vascular clogging with RBCs, we labeled RBCs with Cr-51, injected them IV, and measured their biodistribution. As a positive control, we intentionally aggregated RBCs by incubating them with the monoclonal antibody Ter119, that binds to the glycophorin-A complex on RBCs, along with a secondary antibody that serves as a cross-linker. As seen in Fig. 4c, the positive control (Ter119) causes IV-injected RBCs to massively accumulate in the lungs. By contrast, RH NCs do not change lung uptake or blood levels at all, compared to the negative control (IV-injected RBCs without NCs).

Having demonstrated that RBCs carrying NCs are not retained in the target organ, we next checked that RH RBCs do not adversely affect PAP. This is one of the most sensitive readouts of mechanical occlusion and adhesive abnormality in the pulmonary microvasculature[29]. Using the same injection protocol as in Fig. 4c, we measured PAPs and found that while antibody-aggregated RBCs used as a positive control did increase PAPs, RH nanogels did not at all, compared to the negative control (RBCs without NCs; Fig. 4d, e).

We next showed that RH does not affect oxygenation or alveolar architecture. Using a similar protocol to Fig. 4d, we injected mice either with RH nanogels or RBCs without NCs and found no difference in blood oxygen levels, as measured by pulse oximetry (Fig. 4f). Similarly, injected mice then had their lungs prepared for histology, and hematoxylin-and-eosin (H&E)

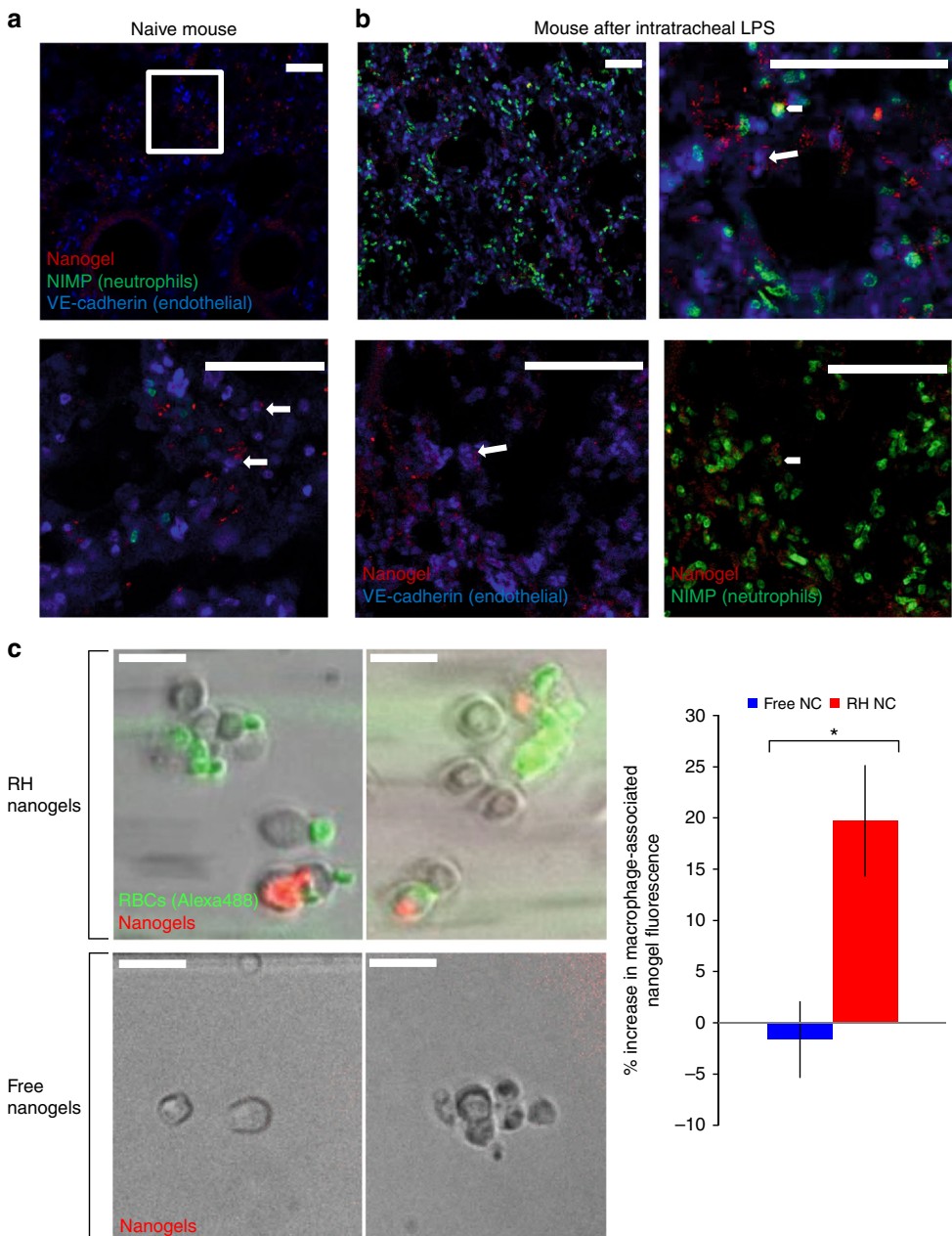

**Fig. 3** RH NCs are taken up by endothelial cells and leukocytes of target organ capillaries. **a** Mice were IV injected with RH rhodamine-conjugated nanogels (NGs), sacrificed 30 min later, and then the lungs were fixed and sectioned. The sections were stained with an endothelial marker (VE-cadherin, blue) and a leukocyte marker (NIMP, green). Rhodamine-NG fluorescence localized to the capillary endothelial cells with small amounts of nanogel signal colocalizing with sparse leukocytes (top row 10×, bottom row 40×). Thick white lines represent scale bars of 100 µm. **b** Mice were given intratracheal LPS to model ARDS prior to RH NCs injection as in **a**. The top left panel is a 10× image, while the top right is a 40× magnification plus digital zoom of the same region. As seen in the top left image, red nanogels are present in an overlapping distribution with both endothelial cells (blue) and leukocytes (green). The two bottom panels are 40× images of another region of the tissue, but displaying only two markers each for ease of viewing. Thick white lines represent scale bars of 100 µm. **c** Macrophages were plated in flow chambers and NGs (labeled red) were introduced, either free or adsorbed onto RBCs (labeled green). As seen in two separate experiments using RH NGs (top two panels), the RH NGs (red) localize to the center of the macrophages, and the RBCs (green) transiently localize on the periphery of the macrophages. In two other experiments using free NGs (bottom two panels), very little NG signal localizes with the macrophages. Scale bars represent 20 µm. In the rightmost panel, we have quantified the gain in macrophage-co-localized red fluorescence (corresponding to nanogels) during the course (10 min) of each of these experiments. Each data point represents mean ± s.e.m ($n = 4$ for free NG, $n = 12$ for RH NGs); *$P < 0.05$, non-paired, two-tailed $t$-test

staining revealed no differences in alveolar architecture between RH nanogels and negative controls (Fig. 4g).

Our final test of safety was to verify that RH does not worsen the existing pathologies. We tested this in our mouse model of ARDS, intratracheal LPS. LPS markedly increased the leukocytes (Fig. 4h, left panel) and edema-related protein in the alveoli (Fig. 4h, right panel), but the addition of RH did not lead to any change in these sensitive inflammatory markers.

**RH works in multiple species**. With the safety of RH verified in mice, we sought to determine if RH worked in larger species. It was possible that RH only worked for the particular RBC and capillary properties found in mice. Therefore, we began by testing the first step of RH, NC adsorption onto RBCs. NC adsorption worked with mouse, rat, pig, and human RBCs, though with a range of adsorption efficiencies (Fig. 5a).

Next, we tested the RH efficiency in vivo in a common large animal model, live pigs. IV injection of RH NCs led to a lung-to-liver ratio of 120 (Fig. 5b, left panel), meaning that the concentration of NCs in the target organ, lungs (#NCs per gram of tissue), was 120× higher than the liver. That lung-to-liver ratio was 17-fold higher than that seen with free NCs; an RH-to-free proportionality similar to the proportionality seen in mice (14.6×,

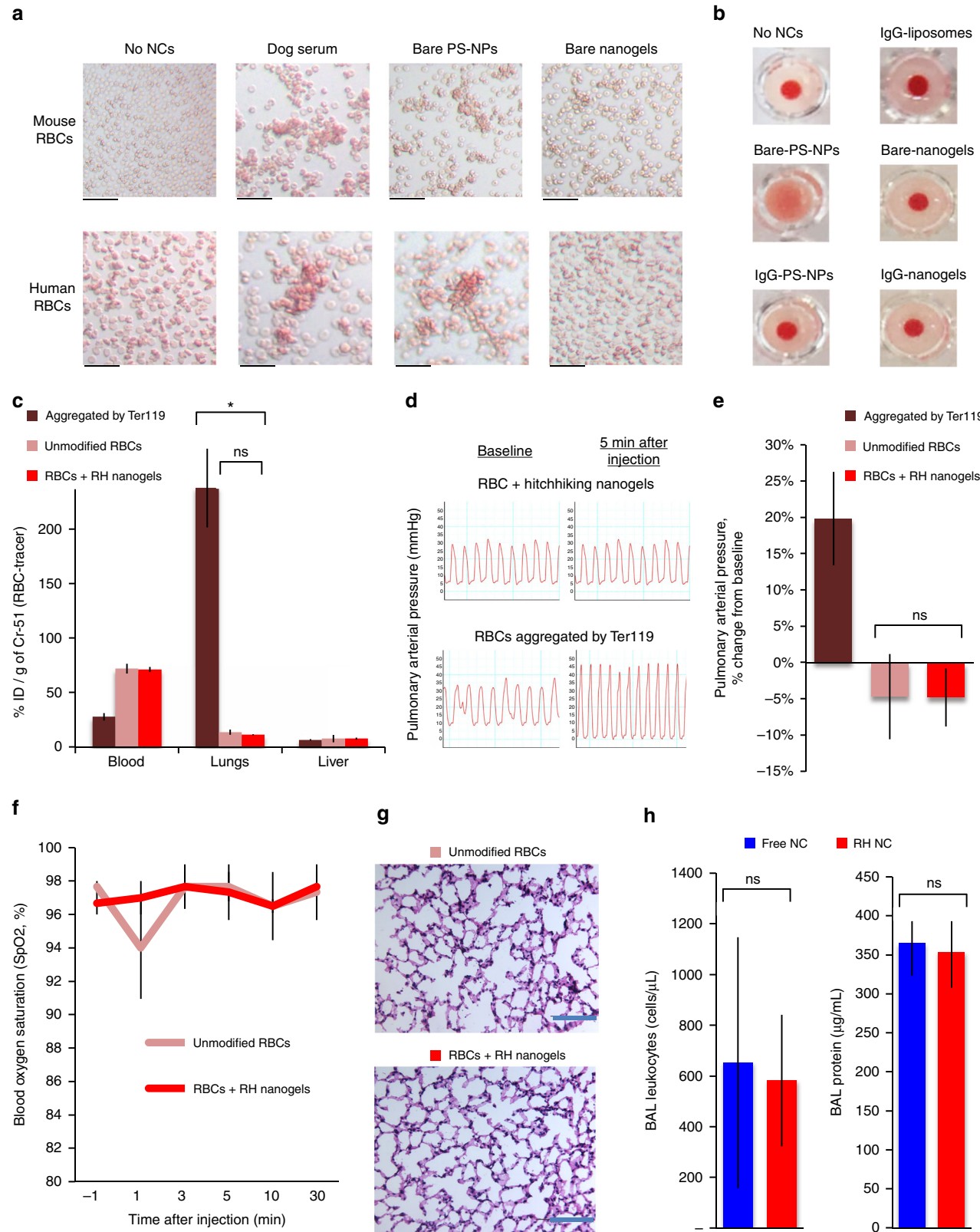

Fig. 2c). Similar augmentation of lung targeting by RH in the pig was demonstrated by the lung-to-blood ratio of 490 (Fig. 5b, right panel). Importantly, as in mice, RH in pigs did not lead to RBCs themselves getting stuck in the lungs, as determined by the biodistribution of Cr-51-labeled RBCs (Fig. 5c). Notably, the data in Fig. 5b, c were from the same individual pigs, with the RH NCs labeled with I-125 accumulating strongly in the lungs (Fig. 5b), while the Cr-51-labeled RBCs did not accumulate in the lungs, showing the RBCs are "dropping off" the NCs in the target organ in the large animal.

To extend the RH data from large animals to humans, we tested the effectiveness of RH in human lungs ex vivo. We obtained fresh human lungs from an organ donor whose lungs were deemed unsuitable for donation due to the presence of capillary leak and pulmonary edema similar to ARDS, the target lung disease for RH. Within 12 h of organ donor death, we endovascularly cannulated subsegmental branches of the pulmonary artery (Fig. 5d, left panel). Into one such artery, we infused both I-125-labeled NCs adsorbed onto RBCs and I-131-labeled NCs that were "free" (not adsorbed onto RBCs), followed by a green tissue dye to mark the infused tissue and prove perfusion to a single lung subsegment (Fig. 5d, right panel). The reason to inject both the RH and free NCs into the same artery is that co-injection into a single vessel controls for differences in quality of endovascular cannulation/vessel isolation, and the fact that ARDS pathology is patchy, which can affect NC uptake[30].

As shown in Fig. 5e's left panel, 41% of the injected dose of RH NCs was deposited in the entire lung. When the tissue was subdivided into 2 mL volumes including only the top 10% best-perfused volumes (judged by green tissue dye intensity; Fig. 5e, right panel), it could be seen that there was 3.7× more deposition of RH NCs than free NCs. Notably, the background lung uptake of free NCs was much higher than we found in mice, which may have been due to the severity of ARDS-like pathology and cold-ischemia time after transplant harvesting, which can cause severe capillary leak and microvascular clots. Nevertheless, RH clearly increased NC delivery to human lungs, in a pathological setting very similar to the goal clinical application of RH in the lungs.

**RH delivers therapeutic cargo to ameliorate a PE model.** Once we established RH's strong delivery without side effects, we next endeavored to show that RH can ameliorate an animal model of disease. We chose a previously established model of PE, in which fibrin micro-clots are prepared in vitro and then injected IV, where they embolize in the lungs[31,32]. As our therapeutic cargo, we chose the FDA-approved thrombolytic enzyme reteplase, which lyses pulmonary emboli on its own, but suffers from off-target side effects in non-lung organs[33].

We began by conjugating reteplase onto nanogels (NGs; Fig. 6a, inset), and then showed that the reteplase-NGs dissolved fibrin clots in vitro, identically to free reteplase (Fig. 6a). We next

adsorbed reteplase-NGs onto RBCs and injected the RH reteplase-NGs into mice (Fig. 6b) and found that, at 1 h, the lung uptake was 250% ID/g (nearly the same as RH IgG-NGs, as in Fig. 2d). Thus, the reteplase-NGs retain full enzymatic activity and RBC-hitchhike strongly to the lungs.

We next tested the RH reteplase-NGs in the PE model. Mice were injected with I-125-labeled fibrin micro-clots, allowing us to measure the PE burden by measuring I-125 levels in the lungs and other organs, a previously validated methodology for measuring PE burden[31,32]. These PE model mice were also injected with either saline, free reteplase-NGs, or RH reteplase-NGs. As seen in Fig. 6c, the amount of clot in the lungs (I-125) is the same for mice receiving either saline or free reteplase NGs, but RH reteplase-NGs show 16 times lower levels of I-125 in the lungs. As would happen with clot dissolution, as opposed to dislodgement of clots, the blood levels of I-125 fibrin/fibrinogen were markedly higher in mice that received RH reteplase-NGs. We quantified the fraction of initial PE (clot in the lung) that dissolved over 1 h, and found that saline and free-reteplase had the same low level of clot dissolution (performed by endogenous mechanisms), while RH-reteplase dissolved nearly 100% of the initial PE (Fig. 6d). Thus, these experiments are proof-of-principle that RH can strongly localize therapeutics to the lungs and thereby ameliorate a severe disease model.

**RH works in diverse target organs.** Having shown that RH safely delivers NCs to the lungs in multiple species, we next wanted to determine if RH could deliver NCs to multiple different target organs. We hypothesized that RH would target NCs to any organ immediately downstream of the injection catheter. Therefore, instead of injecting IV, we injected into arteries feeding the target organs and assayed the uptake into the organ.

The first organ we chose for IA RH was the brain, which we targeted via injection into the right internal carotid artery, which is the smallest artery branch feeding the brain that we could cannulate in a mouse (Fig. 7a, inset). As seen in Fig. 7a's left panel, RH achieved a brain uptake of 11.5% of the injected dose (%ID). More impressively, the brain-to-liver ratio of RH was 14.3, which was 143-fold higher than that for free NCs (see Supplementary Fig. 1). Similarly, the brain-to-blood ratio of RH was 27-fold higher than for free NCs (Fig. 7a, right panel). To determine where in the brain the RH NCs localize, we injected I-125-labeled RH NCs and then prepared slices of the mouse's brain that were imaged with visible light and autoradiography. Figure 7b shows that the right side of the brain receives significantly more delivery than the left, and that the distribution within the brain varies in correlation with defined anatomical substructures. To further localize the RH NC delivery, we performed a similar experiment to autoradiography, but injected RH NCs that were labeled with rhodamine and performed immunofluorescent microscopy. Numerous puncta were seen in

**Fig. 4** Optimized NCs do not cause toxicity during RH. **a** Mouse and human RBCs were mixed either with an agglutinating cross-species serum (dog serum) or NCs and then prepared as a "thin smear" slide. Dog serum causes RBCs to aggregate, as do bare PS-NPs, but nanogels do not. Scale bar represents 30 μm. **b** Round-bottom well assay of RBC aggregation, in which aggregated RBCs form a diffuse haze, while non-aggregated RBCs settle into a tight red dot. **c** RH nanogels, with the RBCs labeled with Cr-51, were prepared as in Fig. 2 and injected into the mice, followed by sacrifice after 30 min and organ Cr-51 signal measurement on a gamma counter. As a positive control, RBCs were intentionally aggregated with the anti-RBC antibody, Ter119, plus a cross-linking secondary antibody. Each data point represents mean ± s.e.m ($n = 3$). **d**, **e** RH nanogels and Ter119-aggregated RBCs were prepared as in **c**, but before sacrifice, the pulmonary artery pressure (PAP) was measured. Five min after injection, Ter119-aggregated RBCs (positive control) had increased the PAP, while RH nanogels had not (quantification in **e**). Each data point represents mean ± s.e.m ($n = 4$). **f**, **g** Mice were injected as in **c** and then had their blood oxygen measured over time (**f**), followed by their sacrifice at 30 min for histology (**g**), showing no difference between RH and RBCs-only. Scale bar represents 100 μm. Each data point represents mean ± s.e.m ($n = 4$). **h** Mice were injected with RH or free nanogels and intratracheally instilled with LPS, followed by measurement of the bronchoalveolar lavage (BAL) levels of leukocytes and protein, both measures of lung inflammation and ARDS. Each data point represents mean ± s.e.m ($n = 3$). *$P < 0.01$, non-paired, two-tailed $t$-test

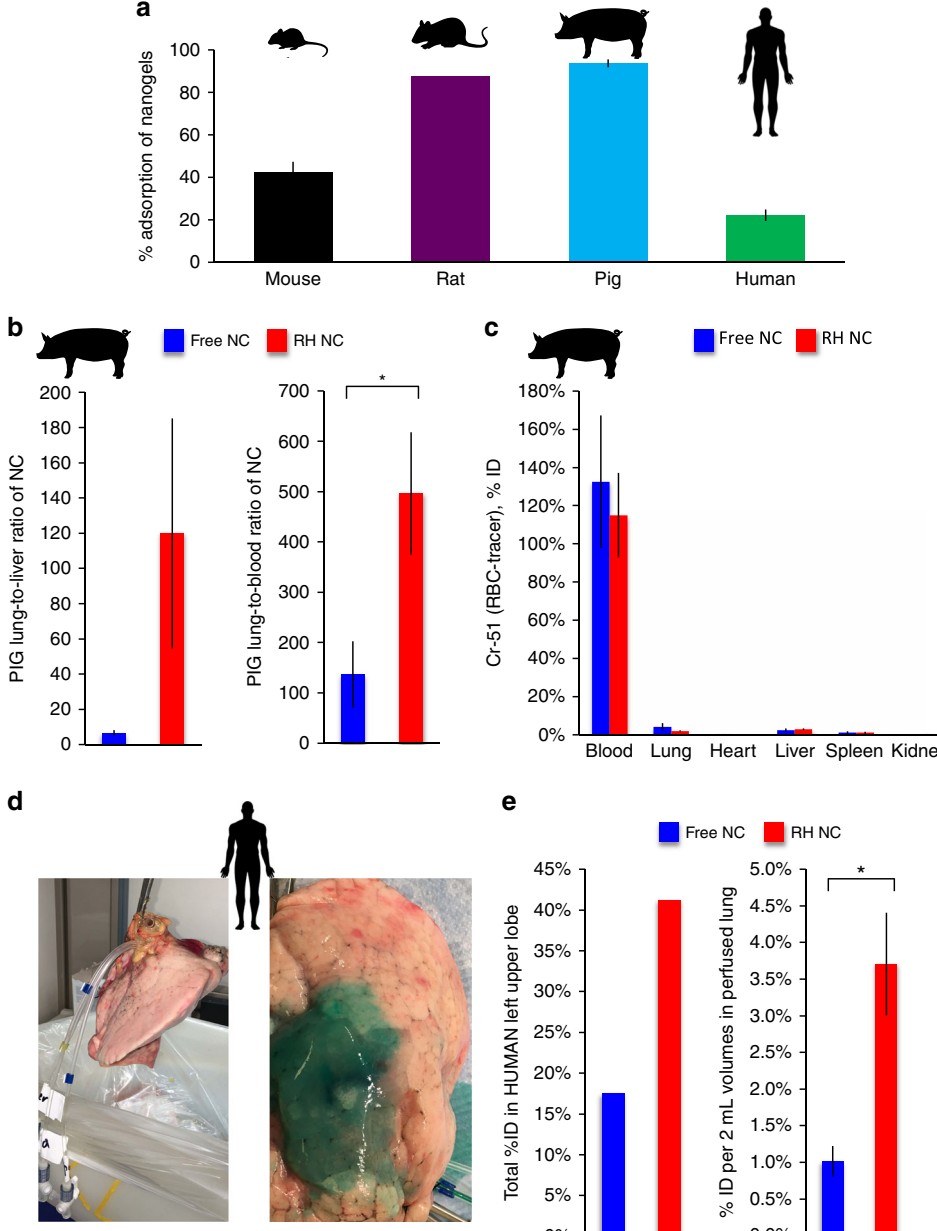

**Fig. 5** RH works in large animals and ex vivo human lungs. **a** Adsorption efficiency of nanogels onto various species' RBCs. Each data point represents mean ± s.e.m (n = 3 replicates per condition). **b** Live pigs were injected with I-125-labeled nanogels that were either free or RH. Plotted are the lung-to-liver and lung-to-blood ratios. Each data point represents mean ± s.e.m (n = 3 pigs per condition). **c** Safety in pigs was assessed by showing that Cr51-labeled RBCs used in the RH experiments of **b** did not get stuck in the lungs. **d, e** Ex vivo human lungs were endovascularly cannulated and then infused through a single artery sequentially with: I-125-labeled RH nanogels; I-131-labeled free nanogels; and a green tissue dye. The percent of the injected dose of the entire lung lobe (**e**, left panel) and within only the well-perfused (green) zone (**e**, right panel, where ten 2-mL volumes of lung were measured and their mean ± s.e.m. listed on the plot). *P < 0.05, two-tailed t-test

the brain, again more on the right side of the brain than left (Fig. 7c). Finally, we performed H&E staining of those brain sections, which showed that the right hemisphere of the brain showed no morphological difference than the left, suggesting no major, acute toxicity (Fig. 7d). Therefore, collectively these data show that an IA injection can safely deliver NCs to the brain at levels markedly above prior technologies.

To generalize our results on IA RH, we next tested RH on a visceral organ, the left kidney, because of ease of cannulation in a mouse. RH NCs had a significantly higher kidney-to-blood ratio than free NCs (Fig. 7e, right panel), though the benefit of RH was not as high as in the brain. Notably, free NCs had very high

baseline levels in the cannulated left kidney (Fig. 7e, left panel), likely because of the ischemic injury that occurs during the ~30 min it takes to tie off and cannulate the left kidney before injection[34]. The brain has strong collateral blood flow between the hemispheres, so such ischemia is less likely to occur. Nonetheless, IA RH clearly improved kidney delivery.

Finally, we aimed to use IA RH to deliver to peripheral tissues. By cannulating the common carotid artery (CCA; not the right internal carotid, which was used for brain experiments), most blood flood goes to the right side of the "face", which we define as all tissues on that side of the head, except for the brain. When injected via the CCA, RH NCs localized in the right "face"

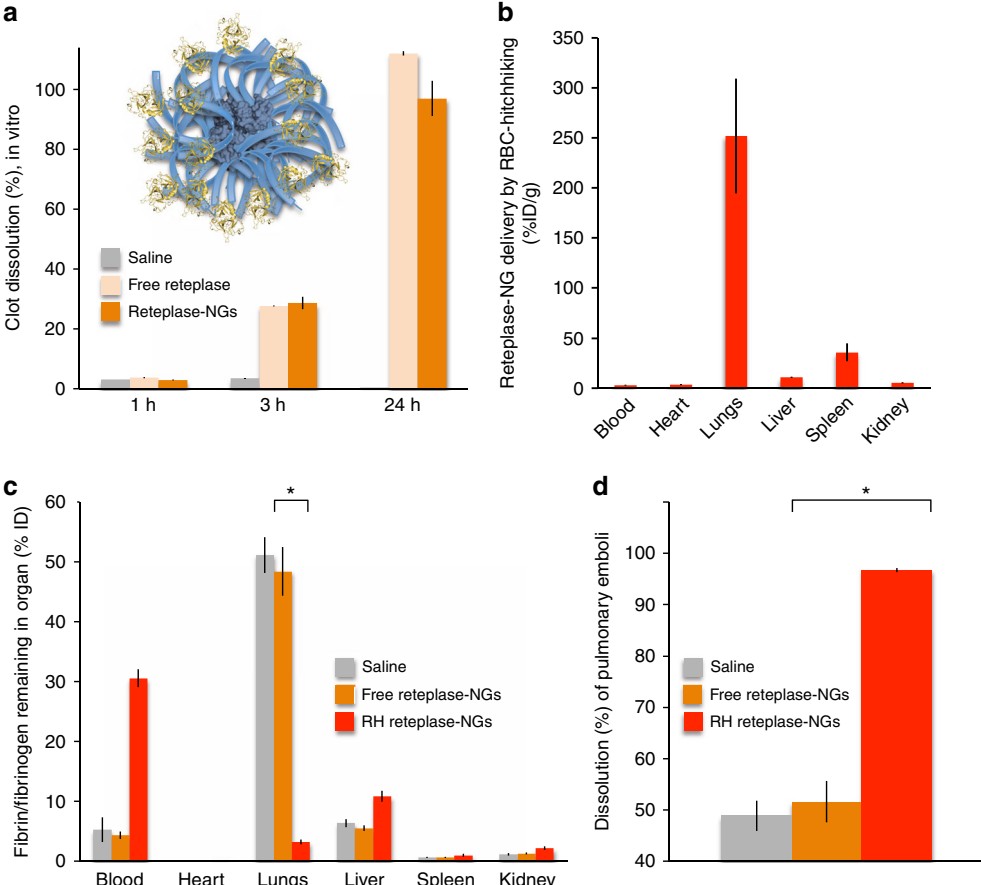

**Fig. 6** RH effectively delivers therapeutic cargo to ameliorate a model of pulmonary embolism. **a** Inset, schematic of nanogels (NGs) with thrombolytic enzyme reteplase (gold) conjugated to the dextran (blue) shell. The bar graph shows the degree to which reteplase-NGs, free reteplase, or saline dissolve preformed fibrin clots in vitro. Reteplase conditions contained 4 nM of reteplase, with reteplase concentration in reteplase-NGs quantified by reteplase-NG coupling efficiency. **b** Biodistribution of RH reteplase-NGs at 1 h after injection, showing high lung uptake. **c** All mice were intravenously injected with ~1–5 micron diameter fibrin clot emboli (I-125-labeled). Mice were also treated with either saline, free reteplase-NGs, or RH reteplase-NGs at 12 micrograms reteplase per mouse. The amount of I-125-labeled fibrin/fibrinogen remaining in each organ at 1 h post injection is quantified. **d** Degree to which initially injected emboli were cleared from the lungs. For all plots in this figure, each data point represents mean ± s.e.m ($n = 4$). *$P < 0.001$, non-paired, two-tailed $t$-test

131-times more than did free NCs, as judged by the ratio of right face to blood levels. Thus, we showed that three diverse organs and tissue types strongly benefit from IA RH, achieving delivery levels far above those of prior technologies.

## Discussion

In this study, we have advanced the original concept of RH from a cell-toxic prototype with modest delivery in mice, to the brink of mapping out the clinical studies. We showed that optimized RH formulations can safely and powerfully target NCs to chosen organs via select placement of intravascular catheters, without the need for an affinity moiety, across multiple animal species including humans.

The magnitude of organ targeting for optimized RH significantly outperforms that of prior affinity-targeting approaches. For example, anti-PECAM antibodies have been used for >20 years to target IV-injected ~100 nm NCs to the lungs with fairly impressive targeting[13]. In the present study, however, anti-PECAM-nanogels had a lung-to-liver ratio 14.6× lower than that achieved by RH-nanogels without an affinity moiety.

RH's advantage over prior technologies is even more pronounced in the context of cerebrovascular targeting. The best brain-targeting affinity moiety published so far is transferrin, which when highly optimized only delivered 1% of the injected dose (%ID) to the brain[3], compared to RH-nanogels, which produced a brain uptake of ~12% ID. These order-of-magnitude advantages of RH should greatly improve the risk–benefit ratios of delivered drugs.

Improved NC delivery and safety of RH resulted from several optimizations, but most notably from the screening of NCs for optimal performance. While all NCs and viral vectors tested worked with RH, it is notable that by far the best performers, liposomes and nanogels, are also the softest (lowest elastic modulus). Of course, different NC types have more differences than just their mechanical deformability. Therefore, to prove optimal NC properties, future studies should use a single NC with tunable properties, including elastic modulus.

The mechanism for RH was originally hypothesized to be the transfer of NCs to capillary endothelium when NC-covered RBCs squeeze through capillaries. We provided the first experimental support for this mechanism by showing via histological analysis that RH NCs are indeed found inside the capillary endothelial cells in the lungs. Surprisingly, we also found that in pathological tissue, RH also delivers NCs to intravascular leukocytes that reside in the capillaries. Consistent with this finding, in vitro

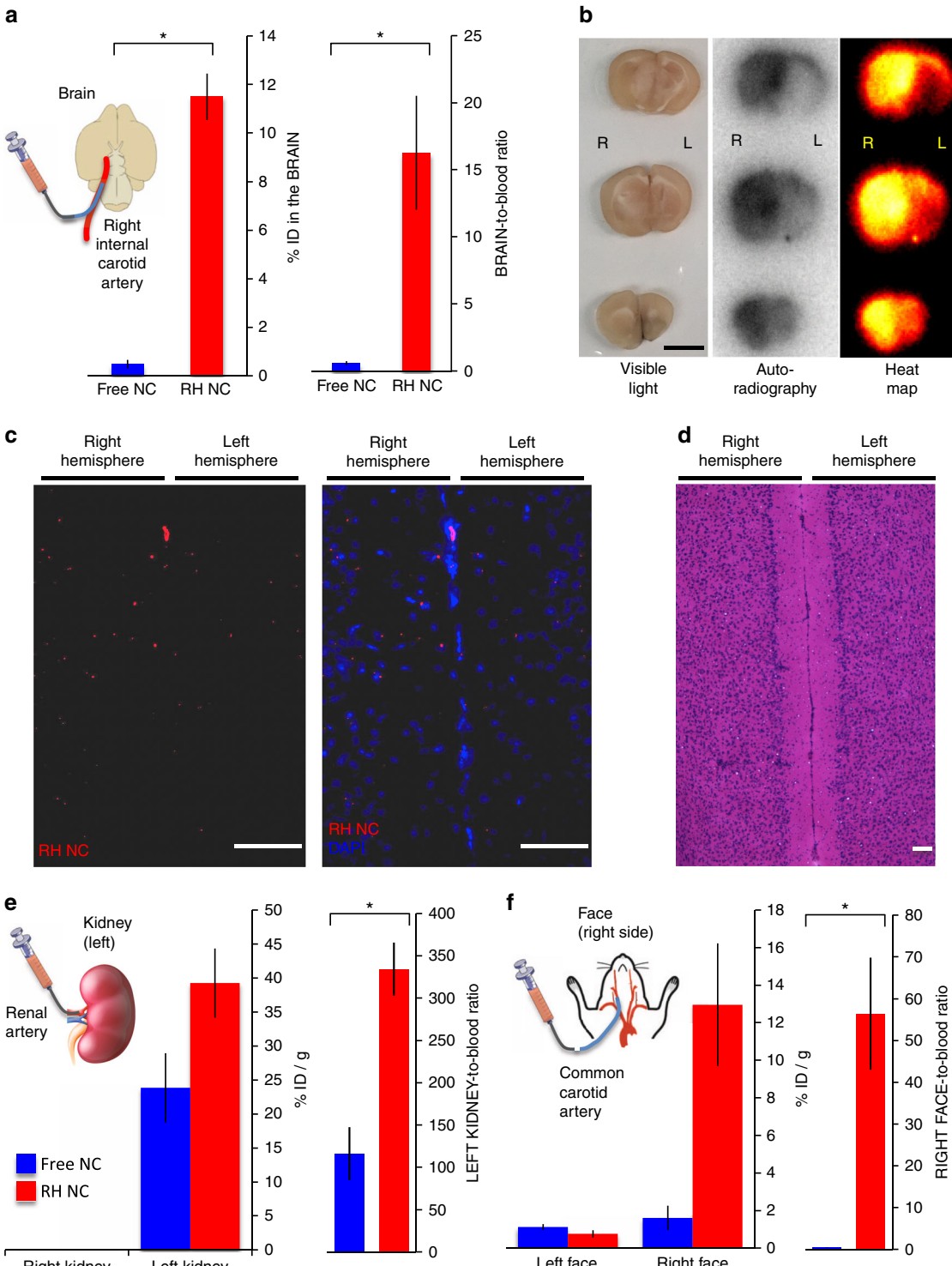

**Fig. 7** IA injection of RH NCs results in high NC uptake in any downstream organ. **a** Mice were anesthetized and had their right internal carotid artery cannulated to deliver the injectate to the brain (inset). Either free (blue) or RH (red) I-125-labeled NCs (here, nanogels (NGs)) were injected. The left panel depicts % injected dose (%ID) in the brain, while the right panel depicts brain/blood ratio. **b** Mice injected as in **a**, followed by sectioning and autoradiography of the brain. Scale bar represents 5 mm. **c** Mice were treated as in **a**, but the injectate was RH rhodamine-labeled NGs. Left panel is rhodamine signal, with more red puncta in the right (targeted) hemisphere. Right panel is rhodamine signal overlaid with DAPI signal. Scale bar represents 100 μm. **d** To demonstrate no gross anatomical damage done to the neurons by RH NGs, brain slices from **c** were H&E stained, showing no difference in morphology between the right and left hemispheres. Scale bar represents 100 μm. **e** Mice were treated as in **a**, but the arterial cannula was in the left kidney. **f** Mice were treated as in **a**, but the arterial cannula was in the common carotid artery, and the right and left sides of the "face" were harvested. Here, a hemi-"face" includes all structures on one side of the head, except for the brain. For all plots in this figure, each data point represents mean ± s.e.m ($n = 3$). *$P < 0.05$, non-paired, two-tailed $t$-test

macrophages in microfluidics arrested flowing RBCs and removed NCs coating them. It is likely that multiple types of intravascular leukocyte take up RH NCs, as eliminating only intravascular macrophages in healthy mice did not affect total lung uptake of RH NCs (Supplementary Figure 2). The lack of effect of eliminating macrophages could mean that other cell types take up RH NCs in the macrophages' absence, or simply that macrophages play a larger role in RH NC uptake in pathological lung tissue than in healthy lungs, consistent with our histology (Fig. 3a vs. Fig. 3b). In the lung pathology, RH is best suited for ARDS, both endothelial cells and local leukocytes are prominent targets for the treatment, so targeting both can have major advantages. Likewise, in the two acute illnesses which cause the most death and disability in the developed world, stroke and heart attack, the endothelium and homing leukocytes play a prominent role in pathology and thus serve as important drug targets[35–37].

The efficient nature and mechanism of RH raises the question of whether RH is recapitulating an evolved mechanism. For example, viruses or other pathogens may adsorb onto RBCs, as we observed with AAV, and then get taken up by the next organ to which they are presented. Notably, the NCs used here are in the same size range as most viruses (e.g., AAV is 25 nm and HIV 120 nm). Indeed, pathogens have been intentionally attached onto RBCs using specific antibodies, and these pathogens are then transferred to the liver[8]. The lung, as the first capillary bed after entry of a NC or pathogen into a vein, may have similarly evolved the function of taking up RBC-adsorbed pathogens as a defense mechanism to protect other downstream organs. Indeed, even though IA RH provides unprecedented delivery to the immediately downstream organ, the lung still gets a disproportionate amount of RH NCs after IA injection (Supplementary Figure 1). Such data intriguingly suggest that perhaps the lung has evolved to be especially adept at receiving RH pathogens and NCs. These data also imply that IA RH should focus on drugs that do not have pulmonary toxicity. Future studies can elucidate the role of RBC adsorption and RH in pathogen uptake into organs.

Turning RH from a potential evolved mechanism into a therapy exploits three size ranges: nanoscale drug carriers; micron-scale RBCs that position the NCs within the vessel lumen; and macro-scale catheters that determine which organ's capillary bed receives delivery. RH's nano-micro-macro synergy provides tremendous target-organ specificity by focusing on delivery of NCs on the "first-pass" of the injectate through an organ. Such a system is now very practical for the most common acute, severe diseases. In severe heart attacks and embolic strokes, standard of care is to insert an IA catheter for removal of the flow blockage, thereby easily allowing IA injection of RH NCs. For the most common severe acute lung disease, ARDS, which accounts for 10% of all intensive care unit (ICU) admissions, only an IV catheter is needed, which all such patients have in place. That means RH is well poised to serve as the delivery modality for the three most important acute, severe illnesses. For these acute applications, there will likely not be time to cross-match the blood, so RH will utilize universal donor blood (O-negative), which is currently used for emergent bleeding standardly in all hospitals. Notably, such intravascular catheters are not practical for most chronic, outpatient-treated diseases, and thus RH development should focus on acute, critical illnesses. However, another major application, which also will utilize directing RH into macro-scale catheters, is delivery into transplanted organs to deliver ischemia-reperfusion-injury drugs. Thus, RH has the potential to provide a drug delivery platform for multiple common diseases that currently lack effective therapies.

## Methods

**Animal and human study protocols**. All animal studies were carried out in strict accordance with Guide for the Care and Use of Laboratory Animals as adopted by National Institute of Health and approved by University of Pennsylvania Institutional Animal Care and Use Committee (IACUC). All studies involving human subjects were approved by the University of Pennsylvania Institutional Review Board. Written informed consent from donors was obtained for the use of blood samples and blood samples were destroyed after the study. Names and personal information about individual participants were not taken. The ex vivo human lungs were donated from the local organ procurement agency, Gift of Life, after they had determined the lungs were not suitable for transplantation into a recipient, and therefore would have been disposed off if they were not used for our study. Gift of Life obtained the relevant permissions for research use of the discarded lungs, and in conjunction with the University of Pennsylvania's Institutional Review Board ensured that all relevant ethical standards were met.

**Synthesis and conjugation of proteins to NCs**. The synthesis, characterization, radiolabeling, and fluorescent labeling of the NCs employed here have all previously been described, including liposomes[30]; nanogels[38]; PLGA-PVA NCs[39]; and albumin-NCs were prepared using electrospraying of an aqueous solution of human serum albumin and PEG-NHS ester (2kD, Sigma)[40]. AAV empty viral capsids were generously provided by Junwei Sun of University of Pennsylvania's Center for Advanced Retinal and Ocular Therapeutics (CAROT). PS-NPs were purchased from Polysciences, Inc. Antibodies used included anti-PECAM (clone Mec13.3; BioLegend Cat # 102502) and anti-ICAM (clone YN1; BioLegend Cat #116108; we also used YN1 produced in our lab from the ATCC hybridoma).

**Blood collection and isolation of RBCs**. Isolation of erythrocytes (RBCs) was performed via the methods previously described[12]. Briefly, whole blood from CJ7BL/6J mice was collected in EDTA; blood from rats as well as human voluntary donors was collected in ~3.2% Na citrate (BD Vacutainer). Blood from pigs was collected in 1× CPDA-1 (Sigma). In addition, whole blood from CJ7BL/6J mice was collected in tubes without any anti-coagulants. Blood was then spun at $1000 \times g$ for 10 min at 4 °C; plasma and buffy coat were removed and discarded. Serum was stored at 4 °C for 3 h until use. Isolated RBCs were washed extensively with 1× Dulbecco's Phosphate Buffered Saline (DPBS), centrifuged ($500 \times g$, 15 min, 4 °C) and the supernatant was discarded. This wash step was repeated for a total of three times.

**Adsorption of NCs to RBCs**. Briefly, RBCs were incubated with either unmodified or IgG-coated NCs at ratios between 200:1 and 3000:1 for 1 h under constant rotation at 4 °C in PBS or different % of serum (0–100%). NC:RBC solution was washed with PBS three times at $100 \times g$ for 8 min to remove unattached NCs.

**Electron microscopy of RBC-adsorbed NCs**. Samples for scanning electron microscopy were fixed in 2.5% glutaraldehyde and 2.0% paraformaldehyde in 1 M cacodylate buffer, ph 7.4, overnight at 4 °C. Red blood cells were allowed to adsorb onto poly-L-lysine-treated coverslips for 1 h, then washed several times in the same buffer. Samples were post-fixed in 2.0% osmium tetroxide for 1 h, washed again in buffer and dehydrated in a graded ethanol series. Samples were treated with several changes of hexamethyldisilazane (HMDS) and allowed to air dry prior to mounting and sputter coating with gold/palladium. SEM images were obtained in an FEI Quanta FEG 250 scanning electron microscope.

**RBC agglutination**. Agglutination of RBCs were performed as previous reported[41]. Briefly, RBC and NC:RBC suspensions at 1% hematocrit were placed onto a U-shaped well plate at 37 °C for 1 h. In addition, RBC and NC:RBC suspensions were placed on a glass slide and observed using a Micromaster microscope (Fisher Scientific) equipped with micro-camera on a 25× objective lens.

**In vitro leukocyte removal of NCs from moving RBCs**. Peritoneal murine macrophages were harvested as described previously[42]. Briefly, 1 mL of 3% Brewer thioglycollate medium was injected IP in C57BL/6 mice. After 4 days, mice were sacrificed via cervical dislocation and 7 mL of sterile cold PBS was injected IP, briefly allowed to permeate the peritoneal cavity, then aspirated with the same syringe. The aspirate was centrifuged for 10 min at $400 \times g$ and the obtained pellet was re-suspended in 400 μL sterile DMEM/F12-10 medium. For addition of macrophages to microfluidics, 100 μL of cell suspension was added per inlet well in 48-well BioFlux (Fluxion Biosciences, San Francisco CA) plates and cells were flowed at 2 dyn/cm$^2$ for 15 min over glass/PDMS microfluidics channels maintained at 37 °C. After an additional 45 min incubation at 37 °C to allow adhesion of macrophages, seeded wells and channels were rinsed three times with 600 μL sterile DMEM/F12 medium via 30 min flow at 2 dyn/cm$^2$, followed by aspiration of remaining medium in inlet and outlet wells. To expose adhered macrophages to nanogel-bearing RBCs, 15 μL RBC/nanogel suspension and 285 μL DMEM/F12 medium were added to inlet wells, prior to flow at 2 dyn/cm$^2$ for 10–15 min with real-time acquisition of fluorescence and brightfield images tracking RBCs (NHS-ester-Alexa 488 label), nanogels (Rhodamine label), and macrophages (brightfield) via BioFlux Montage software.

**IV-injected NCs in mice**. For all mouse experiments, we used C57BL/6 adult mice. Biodistribution of radiotraced NCs and fluorescence imaging of the lungs was performed as previously described[24,30]. For instillation of LPS, the mice were anesthetized with ketamine and xylazine, orotracheally intubated with a 20-gauge angiocatheter, and then LPS was instilled at 1 mg/kg[30]. Pulmonary artery pressures were measured by putting the orotracheally intubated mouse on a ventilator, making a thoracotomy over the left anterior chest, and inserting a pressure sensor into the right ventricle.

**PE model in mice**. We used a thoroughly validated murine model of PE mediated by plasma microemboli (ME) of fibrin clots[31,32]. Fibrin clots were prepared by mixing a solution of 3 mg/mL of bovine fibrinogen with plasminogen, $CaCl_2$, and thrombin (0.5 µM, 20 mM and 0.2 U/mL final concentrations, respectively). To measure fibrinolysis, in vitro tracing doses of 125I-Fg was added. Clots were allowed to mature 20 min at room temperature. For in vitro clot lysis assays, a solution containing the desired amount of thrombolytic was added over the clots. Fibrinolysis was measured by the radioactivity released to the supernatant at the selected times. To induce the in vivo PE model, a suspension of radiolabeled ME (size: 1.5–5 µm) containing 6000 particles/µL was injected through the jugular vein. Free reteplase-NG, RH reteplase-NG, or saline were injected 1 min prior the ME. Animals were sacrificed 1 h later and the residual radioactivity retained in lungs and other organs was measured.

**IA-injected NCs in mice**. The relevant artery was exposed and cannulated with a heparin-coated catheter. The catheter was flushed with PBS immediately before and after NC injection. Thirty minutes after injection, the catheter was flushed again, and the animal sacrificed.

**IV-injected NCs in pigs**. As previously described[43], 2–7-day-old Yorkshire pigs were anesthetized, endotracheally intubated, cannulated in the left internal jugular (for NC injection) and right femoral artery (for blood pressure and blood gas monitoring) and then injected IV with NCs, followed by radiotracer biodistribution measurement as for the mice.

**Ex vivo human lung biodistribution**. Human lungs were obtained after organ harvest from transplant donors whose lungs were in advance deemed unsuitable for transplantation due to radiographic evidence of alveolar filling and low $PaO_2$ to $FiO_2$ (P/F) ratios. The lungs were harvested by the organ procurement team and kept at 4 °C until the experiment, which was done within 12 h of organ harvest. The lungs were first inflated with low pressure oxygen and the main bronchus clamped. Pulmonary artery subsegmental branches were endovascularly cannulated, and then sealed to the artery opening with tissue glue composed of 30% bovine serum albumin (BSA) mixed at 1:1 with glutaraldehyde, thus preventing retrograde efflux of solutions injected into the artery branch. The pulmonary veins were also cannulated to aid with efflux of solutions that flowed through the lungs. The lungs were then perfused with 3% BSA in PBS at 25 cm $H_2O$ pressure. We then injected into the artery, via its sealed catheter, I-125-labeled nanogels that were adsorbed onto human RBCs, and then flowed into the catheter 3% BSA-in-PBS through for 5 min. We then injected into the same artery I-131-labeled nanogels that were not adsorbed onto RBCs. We then ran 3% BSA-in-PBS through for 5 min. Finally, we injected a green tissue dye. The entire lung lobe was then dissected into 2 mL segments, which were analyzed for green dye intensity and measured in a gamma counter. We analyzed the green regions which were perfused by the chosen subsegmental artery separately, and the entire lobe collectively.

**Autoradiography of the mouse brain**. I-125-labeled RH NGs were injected via the right internal carotid artery, followed by sacrifice 30 min later. The brain was extracted, placed in 4% paraformaldehyde overnight, and then the brain was sliced into 1-mm thick coronal sections. Those sections were dried at 4 °C overnight, then placed on a phosphor imaging plate, followed by imaging of the plate on a GE Typhoon Phosphorimager.

**Data availability**. All relevant data are available from the authors upon request.

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

## Acknowledgements

J.S.B. was supported by NIH F32 HL 129665 – 01 and NIH K08 HL138269-01. D.C.P. was supported by NIH T32 HL07915. J.W.M. was supported by NIH T32 HL007971. N. R., R.R., and V.R.M. were supported by NIH U01EB016027. This study was supported in part by NIH via grants to V.R.M. (HL087036, HL090697, and HL121134).

## Author contributions

The manuscript was written through contributions of all authors. All authors have given approval to the final version of the manuscript. J.S.B., D.C.P., J.W.M., O.A.M.C., S.M., and V.R.M. conceived of the concepts of R.H. J.S.B., D.C.P., J.W.M., and O.A.M.C. performed all experiments in collaboration with the other authors. J.S.B., D.C.P., J.W.M., O.A.M.C., and V.R.M. analyzed all data.

## Additional information

**Competing interests:** The following competing financial interests are declared: five of the authors (J.S.B., D.C.P., J.W.M., V.R.M., and S.M.) are listed on a patent application submitted by the University of Pennsylvania, U.S. patent application number 15/722,583, which covers the use of RBC-hitchhiking nanocarriers for the treatment of disease. The remaining authors declare no competing interests.

