## [Peer Review File · Nature Communications]

Reviewers' Comments:

Reviewer #1:

Remarks to the Author:

Targeted delivery of therapeutic payload is a challenging endeavor, and numerous cell carriers have been extensively evaluated in recent years. In this current manuscript used red blood cell hitchhiking loaded/absorbed with nanocarrier for targeted delivery to the lung and the brain. I think that this paper does show a degree of novelty based on the prior concept of nanocarrier targeting with the addition of the particles to red blood cells. The authors do a good job of convincing me that this approach can be translatable, and their validation of the effect of NC uptake across multiple species provides good evidence of the efficacy of future use. The delivery of these NC particles is explored well, and the mechanism of delivery in the lungs is elucidated and well explained. Having said that, I have the following issues regarding this manuscript.

1. The novelty of this work is the weakest point of this manuscript because of the fact that the paper lacks strong justification of the systems functioning in multiple organs outside of the lungs. This targeting of lung tissue has already been shown by this group in 2013 & 2015, and this paper reads more like a re-tune of old methods than a novel approach to target new organs.
2. Most importantly, this paper is missing a disease model where the investigators tested their cell carrier and demonstrated the clinical translatability of this system. For example, they show in Fig. 6 that intra-arterial delivery of RBC-hitchhiked NC resulted in higher NC uptake in the brain. To demonstrate the clinical applicability of this system they should use disease model such as orthotropic brain tumor model, try to deliver blood-brain-barrier non-permeable anti-cancer therapeutic and should demonstrate that RBC-hitchhiked system can deliver in the target organ and can achieve clinically relevant therapeutic efficacy.
3. I would like to see more validation of no harm being done to the RBC's that are coated with the particles. The aggregation assay shown is convincing that the NC addition itself does not cause aggregation, and arterial pressure is also measured and shown not to be increased, but is it known whether this coating affects the transfer of oxygen to or shuttling of carbon dioxide from tissues, or if the NC coating causes damage to arterial or venous walls during transit to the target organ. Experiments designed to address these questions would be critical.
4. One thing that the authors should also consider exploring is whether or not the NC's only pool in the organ downstream of the injection site or whether a portion of the RBC NC's can remain in circulation and traffic to other organs. Demonstrating this does not will bolster the claim that this approach can be targeted to a specific organ.
5. Finally, while the paper demonstrates, using multiple validations, that the NC coated RBC's deliver their targets in the lungs with a much higher ratio than the liver and spleen, and the mechanism of this delivery is explored in depth, the authors do a poor job of convincing the reader that this type of particle delivery carries much effectiveness outside of the lungs. One experiment is done to assay this delivery in the brain, and it was not completed or explained with a depth even close to the prior experiments showing efficacy in the lungs. In fact, the two major organs talked about in the intro, the heart, and brain, are scarcely even studied or referenced in the paper until the last figure, and in this figure, the data is not overall convincing that this new therapy provides a benefit over existing approaches. More validation of the delivery of the NC's is needed in other organs before the title of this paper, as well as the claims it makes in the introduction, can be confidently stated.
6. In Fig.6, the delivery of RBC-hitchhiked NC resulted in higher NC uptake in the brain is exciting as it is extremely difficult to cross the BBB and deliver any therapeutic to the brain. What is mechanism pretending to RBC carrier crossing the BBB without any inflammation? Authors should at least speculate in their discussion. And most importantly, what would be next step to make this carrier only brain or lung-specific?
7. In fig. 6 it would be nice to add another cell type as a negative control to demonstrate the carrier cell specificity towards these target organ.
8. In Fig 3C, what is the cause of the experimental variability between Exp#1 and Exp#2? Please Discuss.

9. The statistical analysis is missing for Fig. 3, 4, 5 and 6.

Reviewer #2:

Remarks to the Author:

The authors have submitted a manuscript building upon their prior work on the utilization of RBCs as nanocarriers. They have presented a well written, well referenced and clear paper with suitable figures and associated supporting documents.

MAJOR POINTS:

1. The authors are using autologous RBCs for this paper. Their data demonstrate effectiveness of this technique and maneuver. However, it is unlikely that non-autologous RBCs could be utilized, and hence, it should be noted this is a patient-selective technique in the manuscript. As such and in the clinical arena, this technique (in its current form) would be a point of care therapy (primarily).
2. The nano-micro-macro concept is quite intriguing and engaging. However, the utility of intraarterial cannulation and administration is an (extremely) invasive technique requiring retrograde (usually) advancement of the device and subsequent therapeutic manipulation/delivery of the payload (drug, stent, etc.) These procedures are fraught with inherent risk and are undertaken in extreme circumstances as espoused by the authors (myocardial infarction, embolic cerebrovascular accident). They can and are done in less urgent/emergent situations, but it still requires a procedure, anesthesia (usually), and associated costs/risks. Again, this is not a limitation that needed to be overcome in the manuscript but it is an acknowledged limitation moving forward. Though 'safe' as an agent in the organ, getting it 'there' will not be without risk in its current form.
3. The paper clearly demonstrates that endothelial cells and leukocytes take up the RBC-hitchhiking NCs. However, if the endothelial cell (or the leukocytes) are the direct target, this technique is effective. However, what if the agent/drug to be delivered needs to get to the parenchymal cells? For example, did type I or type II pneumocytes take up the product? This fact is critical because most drugs/agents do not act only (or ever) on the endothelium to produce their effect(s), but actually on the parenchymal cells within the organ beyond the endothelium. Therefore, the major limitation of this paper is the lack of demonstration of parenchymal deposition/delivery to prove concept definitively and should be documented prior to publication.

Reviewer #3:

Remarks to the Author:

The work by Brenner and colleagues provides an interesting and novel approach for the specific delivery of therapeutic agents (nanocarriers, NC) to targeted organs, based on first adsorbing the NC to erythrocytes (RBCs, red blood cell hitch-hiking, RH). Overall the model experiments are very powerful, and the investigators have been able to demonstrate feasibility in models in mice, pigs, and in an ex-vivo human lung system. Much of the work on mechanisms of uptake and removal of the agents from the RBCs is clear, but it is curious why the role of macrophages has not been investigated. In Figure 3a-b, a stain specific for mouse macrophages should also have been used. In addition, if the mice are first treated with clodronate, to eliminate macrophages, how does this change the pattern of distribution of the injected agents? This does NOT have to be done exhaustively, but it would be important to address this question in at least one of the mouse models.

Other comments:

Page 2: There should be a literature citation to Ehrlich's magic bullet

Page 10: It is not at all clear, based on the data, that "this technique will work "in any organ downstream of the injection catheter." This statement must be modified.

Page 13: "Dan's PLOS" must be fixed.

Page 14: Remove "as" in the second paragraph. Also, remove second comma, 4th paragraph.

Page 15: Please clarify the second line of the top paragraph. What does "let ran" mean?

Response to Reviewers

We appreciate the positive comments from the reviewers, who described our results as “interesting and novel,” “engaging,” “well written,” “clear,” “quite intriguing,” and “very powerful.” Further a reviewer noted that the manuscript does a “good job of convincing me that this approach can be translatable.” We also thank the Reviewers for their constructive feedback, which we are confident has strengthened the manuscript. We have addressed the comments by collecting additional data, performing additional analyses, and clarifying our conclusions. Below is a point-by-point response to the reviewers comments.

Reviewer #1 (Remarks to the Author):

Targeted delivery of therapeutic payload is a challenging endeavor, and numerous cell carriers have been extensively evaluated in recent years. In this current manuscript used red blood cell hitchhiking loaded/absorbed with nanocarrier for targeted delivery to the lung and the brain. I think that this paper does show a degree of novelty based on the prior concept of nanocarrier targeting with the addition of the particles to red blood cells. The authors do a good job of convincing me that this approach can be translatable, and their validation of the effect of NC uptake across multiple species provides good evidence of the efficacy of future use. The delivery of these NC particles is explored well, and the mechanism of delivery in the lungs is elucidated and well explained. Having said that, I have the following issues regarding this manuscript.

1. The novelty of this work is the weakest point of this manuscript because of the fact that the paper lacks strong justification of the systems functioning in multiple organs outside of the lungs. This targeting of lung tissue has already been shown by this group in 2013 & 2015, and this paper reads more like a re-tune of old methods than a novel approach to target new organs.

Response: We agree that our initial submission did not highlight all the significant innovations and novelty of this work. Based on the Reviewer’s suggestions, we have therefore modified the manuscript to highlight those aspects, as listed below. Additionally, we have performed new experiments that further support these innovations. In brief, we have highlighted the following 5 novel aspects of this work:

- First, this manuscript introduces the major innovation of “intra-arterial” RH. While prior intravenous (IV) injection of RH could modestly target one organ (the lungs), our development of intra-arterial RH allows targeting of potentially any organ tissue. Intra-arterial RH will likely find its greatest application in stroke, where use of intra-arterial catheters has in the last 3 years become standard-of-care^{1,2}. Here, we began by showing intra-arterial RH delivers at unprecedented levels to the brain. However, as the reviewer astutely noticed, we had not shown that intra-arterial RH works in other organs. Therefore, in this revision we have added data showing that RH also works for delivery to a visceral organ, the kidney, when injected into the renal artery (Fig 7e). Additionally, we showed that intra-arterial RH works for delivery to peripheral tissues by showing delivery to

the head & neck (extracranial tissues), via injection of RH nanocarriers (NCs) into the common carotid artery (Fig 7f). Thus, for 3 out of the 3 organs in which we tested intra-arterial RH, RH works and delivers unprecedented levels of NCs.

- Second, our newly optimized RH displays effect sizes more than 5x greater than prior RH reports. When RH was first serendipitously discovered using bare polystyrene beads, the effect size (e.g., %ID in the lungs) was much smaller than that obtained with lung-targeting nanoparticles (NPs) that had been reported for >20 years, such as NPs coated with anti-PECAM antibodies^{3,4}. Only by screening numerous NP formulations did we find that two NPs, liposomes and nanogels, performed markedly better than all other screened NPs, and indeed better than anti-PECAM-NPs. By finding which NPs allow high-level organ delivery by RH, we dramatically changed the original RH finding from a curious but minor effect, making RH now a technique with dramatic effect sizes and capable of working with clinically-translatable NPs.
- Third, this work was the first to investigate safety of RH, which turned out to give surprising results. We did not at all expect to find that the NPs used in the first report on RH, bare polystyrene, were quite deleterious to RBCs, causing severe aggregation (see Figure 4). By comparison, the top hits in our NP screen of RH, liposomes and nanogels, caused no such safety issues. This paper was thus pivotal in showing that RH can be safely performed, but only with select NPs, a surprising set of findings crucial for future development.
- Fourth, we are the first to show the mechanism of NP transfer of RH. Previously, it was speculated without evidence that during RH, NPs transfer to the endothelium. Here we used histology to show that speculation was indeed true. Further, we made the unexpected discovery that intravascular leukocytes also participate in the transfer (Figure 3).
- Finally, this manuscript shows that RH works not just in mice, but also rats, large animals (pigs), and even humans. Such generalizability was not easy to

anticipate, since RBC properties differ greatly between species in terms of surface epitopes, stiffness, and the ratio of RBC-diameter to body size. Thus, the novel finding that RH works in many animals suggests a conserved mechanism, and perhaps one that occurs naturally. But most importantly for develop of RH as a medical technology, it works in common large animal models and even ex vivo human lungs.

2. Most importantly, this paper is missing a disease model where the investigators tested their cell carrier and demonstrated the clinical translatability of this system. For example, they show in Fig. 6 that intra-arterial delivery of RBC-hitchhiked NC resulted in higher NC uptake in the brain. To demonstrate the clinical applicability of this system they should use disease model such as orthotropic brain tumor model, try to deliver blood-brain-barrier non-permeable anti-cancer therapeutic and should demonstrate that RBC-hitchhiked system can deliver in the target organ and can achieve clinically relevant therapeutic efficacy.

We very much agree with the reviewers that therapy is the ultimate goal of RH. Therefore, we have demonstrated that RH effectively ameliorates a major disease, pulmonary embolism (PE). PE involves clots that form in deep veins migrating into the lungs, causing shock, hypoxemia, and death. In the US alone, there are 600,000 cases of PE each year, with 60,000 deaths. Thrombolytics, such as the tPA variant reteplase, are used to dissolve the clots, but only a tiny fraction of the reteplase goes to the lungs, with the rest causing off-target bleeding, including life-threatening intracranial bleeds. To show RH may solve that problem, here we used a previously validated model of PE, in which ^{125}I -labeled fibrin clots are injected into the lungs, allowing PE levels to be quantified by measuring ^{125}I levels in the lungs. As shown in Fig 6a, we first made NCs (nanogels) with reteplase conjugated onto the surface, and such reteplase-NCs displayed identical *in vitro* enzymatic activity as unconjugated reteplase. We then measured PE clot dissolution *in vivo* in mice receiving the fibrin-clot PE model, which were also injected either with saline, RH reteplase-NCs, or with free reteplase-NCs. As

seen in Figs 6c & 6d, the lung levels of PE clot burden were no different between mice injected with saline or free reteplase-NCs. In sharp contrast, the mice receiving RH reteplase-NCs had nearly 100% dissolution of the PE clots. Thus, we have demonstrated that RH can effectively deliver a functional therapeutic cargo to a target organ, thereby alleviating local pathology to a much greater extent than delivery by free NCs.

3. I would like to see more validation of no harm being done to the RBC's that are coated with the particles. The aggregation assay shown is convincing that the NC addition itself does not cause aggregation, and arterial pressure is also measured and shown not to be increased, but is it known whether this coating affects the transfer of oxygen to or shuttling of carbon dioxide from tissues, or if the NC coating causes damage to arterial or venous walls during transit to the target organ. Experiments designed to address these questions would be critical.

We wholeheartedly agree that it is of paramount importance to study the safety of RH extensively. Therefore, we have edited the manuscript to highlight our extensive studies of safety here and in another manuscript, and we additionally added new safety experiments according to the reviewers clinically astute impression of where safety issues may arise.

In this manuscript, we highlighted that our top-performing NP, nanogels, do not cause aggregation of RBCs (by two assays; Fig 4a, 4b), do not cause RBCs to get stuck in the organ to which the NPs are transferred (Fig 4c), do not elevate arterial pressures in the target organ (Fig 4d, 4e), and do not cause end-organ increase in capillary leak or leukocyte infiltration (Fig 4h). Additionally, we have recently published a manuscript⁵ focused solely on *in vitro* tests of safety of nanogels adsorbed onto RBCs. There we showed that these NPs do not increase hemolysis, do not increase RBC stiffness, and do increase phosphatidyl serine exposure on RBCs. Notably, all those safety findings and the *in vivo* safety findings in the current manuscript were non-obvious, as the NC used in the RH paper prior to ours (Anselmo, 2013), bare polystyrene, failed terribly in most of these safety tests.

To address the important safety issues the reviewer requested, we have added significantly to our figure covering safety, Figure 4. In Figure 4f, we show that RH does not change pulse the mouse's oxygenation, as measured by pulse oximetry. In Figure 4g, we show via histology that RH does not adversely affect the vascular walls or alveolar architecture. Altogether, our 8 assays of safety in this manuscript strongly suggest that RH does not damage the RBCs or the target organ.

4. One thing that the authors should also consider exploring is whether or not the NC's only pool in the organ downstream of the injection site or whether a portion of the RBC

NC's can remain in circulation and traffic to other organs. Demonstrating this does not will bolster the claim that this approach can be targeted to a specific organ.

This point brings up a very interesting finding. To elucidate this phenomenon, we have added in Supplemental Figure 1, which contains multi-organ biodistributions after injection at 3 different arterial sites (the right internal carotid artery, the left renal artery, and the common carotid artery). As seen in Supplemental Figure 1, compared to free NCs, RH NCs display lower organ uptake in all off-target organs except for the lungs. The lungs have higher uptake with RH. This fits with the reviewers hypothesis that after intra-arterial injection into a target organ, some RBC-NCs pass through without transfer, and transfer to the next capillary bed, which is the lung. We have added to the Discussion section our hypothesis on the mechanism and a discussion of how this may constrain the types of cargo drugs that are optimal for brain-directed RH, as cargo drugs must not have pulmonary toxicity. Nonetheless, while lung delivery is increased after intra-arterial RH, the key finding is that we increase brain delivery by orders of magnitude.

5. Finally, while the paper demonstrates, using multiple validations, that the NC coated RBC's deliver their targets in the lungs with a much higher ratio than the liver and spleen, and the mechanism of this delivery is explored in depth, the authors do a poor job of convincing the reader that this type of particle delivery carries much effectiveness outside of the lungs. One experiment is done to assay this delivery in the brain, and it was not completed or explained with a depth even close to the prior experiments showing efficacy in the lungs. In fact, the two major organs talked about in the intro, the heart, and brain, are scarcely even studied or referenced in the paper until the last figure, and in this figure, the data is not overall convincing that this new therapy provides a benefit over existing approaches. More validation of the delivery of the NC's is needed in other organs before the title of this paper, as well as the claims it makes in the introduction, can be confidently stated.

These are very good points and therefore we have addressed them in multiple ways.

First, we have added in Figures 7e and 7f (shown in comment #1), in which we showed RH works after intra-arterial injection in 3 out of 3 organs tested: the brain, kidney, and the face (extracranial tissues of the head). See Comment #1 above for the figure and explanation.

Second, we have added data that further elucidate delivery to the brain (Fig 6a-6d). First, we investigated whether RHNCs localize uniformly throughout the brain or display regional enrichment. We injected I-125-labeled RHNCs into the right internal carotid artery, and then sliced the mouse's brain to image it with autoradiography (Fig 6b). As shown above, the RHNCs were strongly biased to the right side of the brain (the side injected into), though the left hemisphere receives some, likely due to the high degree of collateral flow that occurs via the Circle of Willis. Additionally, within the right hemisphere it is evident that NCs localize at higher concentrations in a few anatomically defined brain regions, such as the striatum. We next performed a similar experiment with rhodamine-labeled RHNCs (Fig 6c), and found again the right-left asymmetry, and

relatively large puncta, whose size ($\gg 200$ nanometer NCs) is reminiscent of the puncta found in cells in the lungs after RH. Finally, we attempted to determine if RH caused local tissue damage in the brain, by examining H&E sections of the brain (Fig 6d). Since RH NCs deliver much more to the targeted hemisphere (Fig 6b), we would expect more damage to the targeted (right) side. However, we observe no right-left asymmetry in tissue morphology, suggested RHNCs do not damage brain tissue. Similarly, the RH NC brains displayed no differences compared to uninjected mice. While this is only a first test of safety of RH in the brain, it is encouraging, and an initial step towards the in-depth safety testing presented for the lung in Figure 4.

In summary, we have shown that intra-arterial RH is generalizable to most, if not all, organs.

6. In Fig.6, the delivery of RBC-hitchhiked NC resulted in higher NC uptake in the brain is exciting as it is extremely difficult to cross the BBB and deliver any therapeutic to the brain. What is mechanism pretending to RBC carrier crossing the BBB without any inflammation? Authors should at least speculate in their discussion. And most importantly, what would be next step to make this career only brain or lung-specific?

Crossing the BBB is an important and popular goal in targeted drug delivery, because many therapeutics require delivery to the parenchyma (mostly neurons). However, it is not essential to cross the BBB for many diseases and drug cargos. Most importantly, the intended application of RH is to be delivered after mechanical thrombectomy for ischemic stroke. In that situation, we would want to deliver drugs to the endothelium, not to the parenchyma. The endothelium serves as the gatekeeper for the parenchyma, controlling influx of serum proteins and leukocytes, which have a major influence on ischemia-reperfusion injury^{6,7}. Therefore, many groups have used drugs that target the brain's endothelial cells in order to improve outcomes in stroke⁸. Thus, our goal with RH is to deliver drugs to or near the brain's endothelium, not cross the BBB.

To address this comment of the Reviewer, we edited the manuscript so that we ensure there are no suggestions that RH NCs cross the BBB. Instead we focus on the fact that delivery to the brain, tens of minutes after injection, is orders of magnitude higher than that achieved by free drugs.

7. In fig. 6 it would be nice to add another cell type as a negative control to demonstrate the carrier cell specificity towards these target organ.

Using other cell types as carriers for NCs is an interesting idea. It has already been demonstrated for macrophages, neutrophils, and more. Because of those pre-existing studies, and because putting in activated leukocytes into the brain would be deleterious, we did not repeat the difficult carotid-injection experiments with leukocytes.

However, we did show that RH works even with RBCs that are no longer viable: RBC-ghosts. RBC-ghosts are produced by transiently exposing RBCs to hypotonic solution, which causes their internal contents to leak out, with the cells becoming translucent or white. As seen in Supplemental Figure 2, RBC-ghosts perform just as well in RH as living RBCs.

8. In Fig 3C, what is the cause of the experimental variability between Exp#1 and Exp#2? Please Discuss.

There appears to be significant heterogeneity between the individual cells in terms of morphology and phagocytosis, as has previously been reported for macrophages⁹. To address this, we have increased our number of replicates in this experiment (n=12 for RH NCs) to provide a better estimate of the range of responses of this relatively heterogeneous cell type. Our quantification of this (Fig 3c, rightmost panel) shows the RH effect is large and that the phagocytic uptake is clearly different with RH than seen with equimolar amounts of free NCs:

9. The statistical analysis is missing for Fig. 3, 4, 5 and 6.

We have added in the requested analyses.

Reviewer #2 (Remarks to the Author):

The authors have submitted a manuscript building upon their prior work on the utilization of RBCs as nanocarriers. They have presented a well written, well referenced and clear paper with suitable figures and associated supporting documents.

MAJOR POINTS:

1. The authors are using autologous RBCs for this paper. Their data demonstrate effectiveness of this technique and maneuver. However, it is unlikely that non-autologous RBCs could be utilized, and hence, it should be noted this is a patient-selective technique in the manuscript. As such and in the clinical arena, this technique (in its current form) would be a point of care therapy (primarily).

This is an astute point, and speaks to the practical challenges that would face a novel therapy like RH. To address this, we have added to the Discussion section text discussing the need for ABO-compatibility. In brief, for emergency situations, we could use universal donor blood (O-negative), as is done in emergency transfusions for patients presenting to the hospital with life-threatening bleeding. Such emergent transfusions are done routinely in the hospital, so there are always stocks of such blood ready with only minutes advanced warning.

2. The nano-micro-macro concept is quite intriguing and engaging. However, the utility of intraarterial cannulation and administration is an (extremely) invasive technique requiring retrograde (usually) advancement of the device and subsequent therapeutic manipulation/delivery of the payload (drug, stent, etc.) These procedures are fraught with inherent risk and are undertaken in extreme circumstances as espoused by the authors (myocardial infarction, embolic cerebrovascular accident). They can and are done in less urgent/emergent situations, but it still requires a procedure, anesthesia (usually), and associated costs/risks. Again, this is not a limitation that needed to be overcome in the manuscript but it is an acknowledged limitation moving forward. Though 'safe' as an agent in the organ, getting it 'there' will not be without risk in its current form.

We agree with this very perceptive analysis of the practicalities of drug delivery in a real hospital setting. We have therefore, in the Discussion, made it more clear that there are only a few select indications in which RH is likely to be practical, most notably in the treatment of 3 life-threatening diseases which require emergent treatment and already have the intravascular catheters in place: ischemic stroke (after mechanical thrombectomy), myocardial infarction (MI; during percutaneous coronary intervention), and ARDS. While these are only 3 indications, they represent huge numbers of patients and huge numbers of lives lost: 700,000 ischemic strokes/year, 800,000 myocardial infarctions, and 190,000 ARDS cases.

3. The paper clearly demonstrates that endothelial cells and leukocytes take up the RBC-hitchhiking NCs. However, if the endothelial cell (or the leukocytes) are the direct target, this technique is effective. However, what if the agent/drug to be delivered needs to get to the parenchymal cells? For example, did type I or type II pneumocytes take up the product? This fact is critical because most drugs/agents do not act only (or ever) on the endothelium to produce their effect(s), but actually on the parenchymal cells within the organ beyond the endothelium. Therefore, the major limitation of this paper is the lack of demonstration of parenchymal deposition/delivery to prove concept definitively and should be documented prior to publication.

It is certainly true that many diseases and drugs require delivery to the parenchyma, such as delivery to type II pneumocytes for idiopathic pulmonary fibrosis and delivery to neurons for Parkinson's disease. However, the endothelium is also a major drug target in many diseases⁶. Notably, the endothelium plays a major role in the 3 diseases we believe RH is best suited to from a drug-delivery perspective: stroke, MI, and ARDS^{7,8,10}. In stroke and MI, the endothelium serves as the gateway for the influx of serum proteins and leukocytes which exacerbate ischemia-reperfusion-injury. In ARDS, the endothelium is responsible for the capillary leak that causes the alveoli to fill with liquid, thus causing deoxygenation. Thus, targeting the endothelium is indeed one of the major goals for these diseases. We have added these sentiments to the Discussion.

Reviewer #3 (Remarks to the Author):

The work by Brenner and colleagues provides an interesting and novel approach for the specific delivery of therapeutic agents (nanocarriers, NC) to targeted organs, based on first adsorbing the NC to erythrocytes (RBCs, red blood cell hitch-hiking, RH). Overall the model experiments are very powerful, and the investigators have been able to demonstrate feasibility in models in mice, pigs, and in an ex-vivo human lung system. Much of the work on mechanisms of uptake and removal of the agents from the RBCs is clear, but it is curious why the role of macrophages has not been investigated. In Figure 3a-b, a stain specific for mouse macrophages should also have been used. In addition, if the mice are first treated with clodronate, to eliminate macrophages, how does this change the pattern of distribution of the injected agents? This does NOT have to be done exhaustively, but it would be important to address this question in at least one of the mouse models.

This is a very good point. Since intravascular leukocytes were a surprise player in the mechanism of RH, we agree that further investigation is warranted. We therefore pursued the reviewer's suggested experiment with clodronate. To eliminate

intravascular macrophages, we injected into each mouse 280 uL of 0.5mg/mL clodronate liposomes 48 hours before IV injection of RH NCs, a scheme which we have previously shown eliminates intravascular macrophages. As seen below, clodronate had no effect on the biodistribution. This suggests that macrophages may play a small role in the total lung uptake in a healthy animal. In retrospect, that may have been expected, since healthy rodents have very low numbers of pulmonary intravascular macrophages (PIMs; ¹¹), but PIMs accumulate in ARDS lungs, such as those we analyzed with histology in Figure 3. Therefore, we attempted giving clodronate and then intratracheal LPS, but the mice all died and we ran out of time while optimizing the dosing. It seems likely that the leukocytes we stained in Figure 3 are actually “marginated” neutrophils, since the lungs have a very high concentration of such resident intra-vascular neutrophils both in health and disease ¹². However, given our in vitro results in Figure 3, it is likely that any “professional phagocyte,” whether it’s a neutrophil or macrophage, is able to pull NCs off RBCs.

Other comments:

Page 2: There should be a literature citation to Ehrlich’s magic bullet

Agreed and done.

Page 10: It is not at all clear, based on the data, that “this technique will work “in any organ downstream of the injection catheter.” This statement must be modified.

This is an excellent point. Our original manuscript only tried two organs: delivery to the lungs via IV catheter and delivery to the brain via intra-arterial (carotid) catheter. To further demonstrate the generalizability, we have added Figure 7 data which shows that intra-arterial RH works in the kidney and in the face. The facts that intra-arterial RH works for 3 different organs, and intravenous RH works for the lung, is at the very least strongly suggestive that it will work for many or most organs.

Page 13: “Dan’s PLOS” must be fixed.

Agreed and done.

Page 14: Remove “as” in the second paragraph. Also, remove second comma, 4th paragraph.

Agreed and done.

Page 15: Please clarify the second line of the top paragraph. What does “let ran” mean?

Agreed and done.

Thank you all for your wonderful feedback, which led to a remarkable improvement in the manuscript.

References

1. Albers, G. W. et al. Thrombectomy for Stroke at 6 to 16 Hours with Selection by Perfusion Imaging. *N. Engl. J. Med.* **378**, 708–718 (2018).
2. Saber, H., Rajah, G. B., Kherallah, R. Y., Jadhav, A. P. & Narayanan, S.
Comparison of the efficacy and safety of thrombectomy devices in acute stroke : a network meta-analysis of randomized trials. *J. Neurointerv. Surg.* (2017).
doi:10.1136/neurintsurg-2017-013544
3. Anselmo, A. C. et al. Delivering nanoparticles to lungs while avoiding liver and spleen through adsorption on red blood cells. *ACS Nano* **7**, 11129–11137 (2013).
4. Muzykantov, V. R. et al. Streptavidin facilitates internalization and pulmonary targeting of an anti-endothelial cell antibody (platelet-endothelial cell adhesion molecule 1): a strategy for vascular immunotargeting of drugs. *Proc. Natl. Acad. Sci. U. S. A.* **96**, 2379–2384 (1999).
5. Pan, D. C. et al. Nanoparticle Properties Modulate Their Attachment and Effect on Carrier Red Blood Cells. *Sci. Rep.* **8**, 1615 (2018).
6. Sieve, I., Münster-Kühnel, A. K. & Hilfiker-Kleiner, D. Regulation and function of endothelial glycocalyx layer in vascular diseases. *Vascul. Pharmacol.* **100**, 26–33 (2018).
7. Yang, Q., He, G.-W., Underwood, M. J. & Yu, C.-M. Cellular and molecular

mechanisms of endothelial ischemia/reperfusion injury: perspectives and implications for postischemic myocardial protection. *Am. J. Transl. Res.* **8**, 765–777 (2016).

8. Tuttolomondo, A. *et al.* Inflammation as a therapeutic target in acute ischemic stroke treatment. *Curr. Top. Med. Chem.* **9**, 1240–1260 (2009).

9. Bain, C. C. & Jenkins, S. J. The biology of serous cavity macrophages. *Cell. Immunol.* (2018). doi:10.1016/j.cellimm.2018.01.003

10. Wang, T. *et al.* Endothelial cell signaling and ventilator-induced lung injury: molecular mechanisms, genomic analyses, and therapeutic targets. *Am. J. Physiol. Lung Cell. Mol. Physiol.* **312**, L452–L476 (2017).

11. Schneberger, D., Aharonson-Raz, K. & Singh, B. Pulmonary intravascular macrophages and lung health: what are we missing? *Am. J. Physiol. Lung Cell. Mol. Physiol.* **302**, L498–503 (2012).

12. Doerschuk, C. M. *et al.* Leukocyte and platelet margination within microvasculature of rabbit lungs. *J. Appl. Physiol.* **68**, 1956–1961 (1990).

Reviewers' Comments:

Reviewer #1:

Remarks to the Author:

This is a revised manuscript where authors demonstrated that red blood cell-hitchhiking (RH) could be used to efficiently deliver the therapeutic cargo to the target organs. This manuscript is now much-improved manuscript, and the author should be commended for their effort to address the majority of the reviewers' critics. Notably, adding a disease model pulmonary embolism now demonstrated the translational value of this approach. However, it is clear how the effectiveness of such approach was quantified in Fig. 6d. It would be nice to add a reference to their method section. Also, immunohistochemical analysis of animal lungs demonstrating the clearance of the emboli would significantly strengthen their study.

Reviewer #3:

Remarks to the Author:

The investigators have produced a first-rate manuscript that is for the most part now acceptable for publication. A few issues need to be resolved.

First, the response to my question concerning macrophages and clodronate is fine, but it (and the figure) MUST be included in the paper, either as a regular figure, or as a supplementary figure. Other readers will have the same question about macrophages, and the figure is needed to address these questions.

Page 7, 4th paragraph: "F" should be replaced by "4h"

Page 8, third paragraph: should read: "more deposition of RH NCs than free NCs."

Page 9 last paragraph: should read "than for free NCs (Fig. 7a, right panel)"

The discussion would be improved considerably if there is at least one reference to each figure

Page 14, line 4: fix "instilled as at "

Page 15, third paragraph: should read : "The brain was extracted and placed"

Response to Reviewers

We appreciate the positive comments from the reviewers, such as Reviewer #1 saying the authors should be “commended” for extensive improvements, and Reviewer #2 saying that it is now “a first-rate manuscript.” We also thank the Reviewers for their constructive feedback, which we are confident has strengthened the manuscript. Below is a point-by-point response to the reviewers’ constructive comments.

Reviewer #1 (Remarks to the Author):

This manuscript is now much-improved manuscript, and the author should be commended for their effort to address the majority of the reviewers' critics. Notably, adding a disease model pulmonary embolism now demonstrated the translational value of this approach. However, it is clear how the effectiveness of such approach was quantified in Fig. 6d. It would be nice to add a reference to their method section. Also, immunohistochemical analysis of animal lungs demonstrating the clearance of the emboli would significantly strengthen their study.

Response:

- We have added in references to two papers which used the exact same pulmonary embolism (PE) model we employ. Those two papers also show that the radiotracing method we used to measure PE burden correlates with immunohistochemistry (IHC) in this model. We did not add in an IHC figure because we did not have enough time to complete that experiment. Additionally, the radiotracing is far more sensitive and quantifiable than IHC, as shown in the two papers we referenced.

Reviewer #3 (Remarks to the Author):

The investigators have produced a first-rate manuscript that is for the most part now acceptable for publication. A few issues need to be resolved. First, the response to my question concerning macrophages and clodronate is fine, but it (and the figure) MUST be included in the paper, either as a regular figure, or as a supplementary figure. Other readers will have the same question about macrophages, and the figure is needed to address these questions..

Response:

- We have added in a new Supplemental Figure #2, as shown below:

Supplemental Figure 2. Effect of pre-treatment with clodronate-liposomes on the biodistribution of RBC-hitchhiking nanocarriers. Mice were given either vehicle (PBS) or 280 μ L of 0.5mg/mL clodronate liposomes 48 hours before IV injection of RBC-hitchhiking nanocarriers (RH NCs), a scheme which we have previously shown eliminates intravascular macrophages. **a**, Biodistribution of the RH NCs 30 minutes after injection. red = naïve mice, red/white-checked = clodronate mice. **b**, Lung-to-liver ratios from the mice in **a**. **c**, Lung-to-blood ratios. ns = non-significant.